# OMI total bromine monoxide (OMBRO) data product: Algorithm, retrieval and measurement comparisons

Raid M. Suleiman[1], Kelly Chance[1], Xiong Liu[1], Gonzalo González Abad[1], Thomas P. Kurosu[2], Francois Hendrick[3], and Nicolas Theys[3]

[1]Harvard-Smithsonian Center for Astrophysics, Cambridge, MA, USA
[2]Jet Propulsion Laboratory, California Institute of Technology, Pasadena, CA, USA
[3]Royal Belgian Institute for Space Aeronomy, Brussels, Belgium

*Correspondence to*: Raid M. Suleiman (rsuleiman@cfa.harvard.edu)

**Abstract.** This paper presents the retrieval algorithm for the operational Ozone Monitoring Instrument (OMI) total bromine monoxide (BrO) data product (OMBRO) developed at the Smithsonian Astrophysical Observatory (SAO), and shows comparisons with correlative measurements and retrieval results. The algorithm is based on direct nonlinear least squares fitting of radiances from the spectral range 319.0-347.5 nm. Radiances are modeled from the solar irradiance, attenuated by contributions from BrO and interfering gases, and including rotational Raman scattering, additive and multiplicative closure polynomials, correction for Nyquist undersampling, and the average fitting residual spectrum. The retrieval uses albedo and wavelength-dependent air mass factors (AMFs), which have been pre-computed using a single mostly stratospheric BrO profile. The BrO cross sections are multiplied by the wavelength-dependent AMFs before fitting so that the vertical column densities (VCDs) are retrieved directly. The fitting uncertainties of BrO VCDs typically vary between 4 and $7\times10^{12}$ molecules cm$^{-2}$ (~10-20% of the measured BrO VCDs). Additional fitting uncertainties can be caused by the interferences from $O_2$-$O_2$, and $H_2CO$ and their correlation with BrO. AMF uncertainties are estimated to be around 10% with the used single stratospheric only BrO profile. However, under conditions of high tropospheric concentrations, AMFs errors due to this assumption of profile can be as high as 50%.

The retrievals agree well with GOME-2 observations at simultaneous nadir overpasses and with ground-based zenith-sky measurements at Harestua, Norway, with mean biases less than -$0.22\pm1.13\times10^{13}$ molecules cm$^{-2}$ and $0.12\pm0.76\times10^{13}$ molecules cm$^{-2}$, respectively. Global distribution and seasonal variation of OMI BrO are generally consistent with previous satellite observations. Finally, we confirm the capacity of OMBRO retrievals to observe enhancements of

BrO over the U.S. Great Salt Lake despite the current retrieval set up considering a stratospheric profile in the AMF calculations. OMBRO retrievals also show significant BrO enhancements from the eruption of the Eyjafjallajökull volcano, although the BrO retrievals are affected under high $SO_2$ loading conditions by the sub-optimum choice of $SO_2$ cross sections.

## 1 Introduction

Bromine monoxide (BrO) is a halogen oxide, predominantly located in the stratosphere and upper troposphere where, like chlorine monoxide (ClO), it is a catalytic element in the destruction of stratospheric ozone (von Glasow *et al*., 2004; Salawitch *et al*., 2005), but with higher efficiency
per molecule. Sources of tropospheric BrO include bromine release ("explosions") during the Polar Spring (Hausmann and Platt, 1994; Hollwedel *et al*., 2004; Simpson *et al.*, 2007; Begoin *et al*., 2010; Salawitch *et al*., 2010; Abbatt, *et al*., 2012; Blechschmidt *et al*., 2016), volcanic eruptions (Bobrowski *et al.*, 2003; Chance, 2006; Theys *et al.*, 2009;), salt lakes (Hebestreit, *et al.*, 1999; Hörmann *et al*. 2016) and stratospheric transport (Salawitch *et al*., 2010).    Global BrO
measurements from space were first proposed for the Scanning Imaging Absorption Spectrometer for Atmospheric Cartography (SCIAMACHY) instrument (Chance *et al*., 1991) and were first demonstrated with Global Ozone Monitoring Experiment (GOME-1) measurements (Chance, 1998; Platt and Wagner, 1998; Richter *et al*., 1998, Hegels *et al*., 1998), and since with SCIAMACHY nadir (Kühl *et al*., 2008), Global Ozone Monitoring Experiment 2 (GOME-2)
measurements (Theys *et al*., 2011; Toyota *et al*., 2011) and TROPOMI (Seo, *et al.*, 2018). Initial observations of BrO by OMI were first reported by Kurosu *et. al*. (2004). Polar Spring BrO enhancements are known to be associated with boundary layer $O_3$ depletion (Hausmann and Platt, 1994; von Glasow *et al*., 2004; Salawitch *et al*., 2005; Simpson *et al.*, 2007; Salawitch *et al*., 2010; Abbatt, *et al*., 2012). OMI measurements of BrO have been used together with chemical and
dynamical modeling to investigate stratospheric versus tropospheric enhancements of atmospheric BrO at high northern latitudes (Salawitch *et al*., 2010). OMI BrO retrieval using the Differential Optical Absorption Spectroscopy (DOAS) method has been used to study the seasonal variations of tropospheric bromine monoxide over the Rann of Kutch salt marsh (Hörmann *et al*. 2016). The Arctic Research of the Composition of the Troposphere from Aircraft and Satellites (ARCTAS)
campaign (Choi *et al*., 2012) found consistency between BrO column densities calculated from

Chemical Ionization Mass Spectrometer (CIMS) measurements with the tropospheric BrO columns derived from OMI using our operational retrieval algorithm. BrO has been observed from the ground in Harestua, Norway (Hendrick *et al.*, 2007), Lauder, New Zealand (Schofield *et al.*, 2004a, 2004b), Antarctica (Schofield *et al.*, 2006), Utqiagvik (Barrow), Alaska (Liao et al., 2011; Frieß et al., 2011; Liao et al., 2012; Sihler *et al.*, 2012; Peterson *et al.*, 2016), Eureka, Canada (Zhao *et al.*, 2015), Summit, Greenland (Stutz *et al.*, 2011) and the Arctic Ocean (Burd *et al.*, 2017).

Enhancement of BrO in the vicinity of salt lakes like the Dead Sea and the Great Salt Lake have been observed from ground-based measurements (Hebestreit *et al.*, 1999; Matveev *et al.,* 2001; Stutz *et al.*, 2002; Tas et al., 2005; Holla *et al.,* 2015). The active bromine compound release is due to the reaction between atmospheric oxidants with salt reservoirs. Satellite observation of salt lake BrO was first reported over the Great Salt Lake and the Dead Sea from OMI (Chance, 2006; Hörmann *et al.* 2016). Seasonal variations of tropospheric BrO over the Rann of Kutch salt marsh have been observed using OMI from an independent research BrO product (Hörmann *et al.* 2016). Bobrowski *et al.* (2003) made the first ground-based observations of BrO and $SO_2$ abundances in the plume of the Soufrière Hills volcano (Montserrat) by multi-axis DOAS (MAX-DOAS). BrO and $SO_2$ abundances as functions of the distance from the source were measured by MAX-DOAS in the volcanic plumes of Mt. Etna in Sicily, Italy and Villarica in Chile (Bobrowski *et al.*, 2007). The $BrO/SO_2$ ratio in the plume of Nyiragongo and Etna was also studied (Bobrowski *et al.*, 2015). The first volcanic BrO measured from space was from the Ambrym volcano, measured by OMI (Chance, 2006). Theys *et al.* (2009) reported on GOME-2 detection of volcanic BrO emission after the Kasatochi eruption. Hörmann *et al.* (2013) examined GOME-2 observations of BrO slant column densities (SCDs) in the vicinity of volcanic plumes; it showed clear enhancements of BrO in ~1/4 of the volcanos, and revealed large spatial differences in $BrO/SO_2$ ratios.

The purpose of this paper is to describe the OMI BrO operational algorithm and the data product, compare it with ground-based and other satellite measurements and analyze its spatiotemporal characteristics. This paper is organized as follows: Section 2 describes the OMI instrument and the data product. Section 3 gives a detailed description of the operational algorithm including algorithm and product history, spectral fitting, AMF calculations, destriping, and fitting

uncertainties. Section 4 presents results and discussion including comparison with GOME-2 and ground-based zenith-sky measurements at Harestua, Norway, global distribution, seasonality, enhanced BrO from the U.S. Great Salt Lake and Iceland's Eyjafjallajökull volcano. Section 5 concludes this study.

## 2 OMI instrument and OMBRO data product

### 2.1 OMI instrument

OMI was launched on the NASA Earth Observing System (EOS) Aura satellite into a sun-synchronous orbit on 15 July 2004. It is a push-broom imaging spectrometer that observes solar backscattered radiation in the visible and ultraviolet from 270-500 nm in three channels (UV1: 270-310 nm, UV2: 310-365 nm, visible: 350-500 nm) at spectral resolution of 0.42-0.63 nm and spatial resolution in the normal (global sampling) mode ranging from $13\times24$ $km^2$ at direct nadir to about $28\times150$ $km^2$ at the swath edges. The global mode (GM) has 60 ground pixels with a total cross-track swath of 2600 km.

Since June 2007, certain cross-track positions of OMI data have been affected by the row anomaly (http://projects.knmi.nl/omi/research/product/rowanomaly-background.php): some loose thermal insulating material likely appeared in front of the instrument's entrance slit, which can block and scatter the light thus causing errors in level 1b data and subsequently the level 2 retrievals (Kroon *et al.*, 2011). Initially, the row anomaly only affected a few positions and the effect was small. But since January 2009, the anomaly has become more serious, spreading to ~1/3 of the positions and retrievals at those positions are not recommended for scientific use. A flagging field has been introduced in the OMI level 1b data to indicate whether an OMI pixel is affected by this instrument anomaly.

OMI measures $O_3$ and other trace gases, aerosols, clouds, and surface properties. Products developed at the SAO include operational BrO, chlorine dioxide (OClO), and formaldehyde ($H_2CO$; González Abad *et al*., 2015) that are archived at NASA Goddard Earth Sciences (GES) Data and Information Services Center (DISC), and offline ("pre-operational") $O_3$ profile and tropospheric $O_3$ (Liu *et al*., 2010; Huang *et al.*, 2017, 2018), glyoxal ($C_2H_2O_2$) (Chan Miller *et al*.,

2014, 2016) and water vapor ($H_2O$) (Wang *et al*., 2014, 2016) that are available at the Aura validation data center (AVDC). All the products except for the $O_3$ profile product are produced using nonlinear least-squares (NLLS) fitting methods based on those previously developed at the SAO for the analysis of measurements from the GOME (now GOME-1) (Chance, 1998; Chance, *et al*., 2000) and SCIAMACHY instruments (Burrows and Chance, 1991; Chance *et al*., 1991; Martin *et al.*, 2006).

## 2.2 OMBRO data product

The current operational BrO product, OMBRO version 3, contains BrO vertical column densities (VCDs), slant column densities (SCDs), effective air mass factors (AMFs) and ancillary information retrieved from calibrated OMI radiance and irradiance spectra. Each BrO product file contains a single orbit of data, from pole to pole, for the sunlit portion of the orbit. The data product from 26 August 2004 through the present is available at GES DISC. Data used in this study cover the period from 1 January 2005 to 31 December 2014.

## 3 Retrieval algorithm

### 3.1 Algorithm and product history

OMBRO Version 1.0 was released on 1 February 2007, based on a spectral fitting window of 338–357 nm. Version 2.0 was released on 13 April 2008. It included major adjustments for Collection 3 Level 1b data, improved destriping measures, change of the fitting window to 340–357.5 nm, improvements to radiance wavelength calibration, and several improvements for processing near-real-time data. In both Versions 1 and 2, total BrO VCDs were retrieved in two steps: first performing spectral fitting using the basic optical absorption spectroscopy (BOAS) method to derive SCDs from OMI radiance spectra, and then converting from SCDs to VCDs by dividing AMFs. This is similar to current SAO $H_2CO$, $H_2O$ and $C_2H_2O_2$ as mentioned previously. The latest Version 3.0.5, released on 28 April 2011, includes major algorithm changes: the fitting window was moved to 319.0–347.5 nm, and BrO cross sections are multiplied by wavelength-dependent AMFs, which are a function of albedo, before fitting, for a direct retrieval of BrO VCDs. SCDs are similarly retrieved in a separate step by fitting BrO cross sections that have not been multiplied with wavelength-dependent AMFs, and an effective AMF = SCD/VCD is computed. Diagnostic

cloud information from the OMCLDO2 product (Acarreta *et al.*, 2004) was added, and the row-anomaly indicating flags were carried over from the level 1b product. We recommend not to use pixels affected by the row anomaly despite being processed by the retrieval algorithm.

The current algorithm is described in detail in the rest of this section, with spectral fitting in Section 3.2, AMF calculation prior to spectral fitting in Section 3.3, post-processing de-stripping to remove cross-track dependent biases in Section 3.4, and fitting uncertainties and error estimates in Section 3.5.

### 3.2 Spectral fitting

Most aspects of the algorithm physics for the direct fitting of radiances by the BOAS method were developed previously at SAO for analysis of GOME and SCIAMACHY satellite spectra (Chance, 1998, Chance *et al.*, 2000, OMI, 2002; Martin *et al.*, 2006) and in the various algorithm descriptions of other SAO OMI products (Wang *et al.*, 2014; Chan Miller *et al.*, 2014; Gonzalez Abad *et al.*, 2015).

The spectral fitting in the SAO OMI BrO retrieval is based on a Gauss-Newton NLLS fitting procedure, the CERN ELSUNC procedure (Lindström and Wedin, 1987), which provides for bounded NLLS fitting. Processing begins with wavelength calibration for both irradiance and radiance. In each case the wavelength registration for the selected fitting window is determined
independently for each cross-track position by cross-correlation of OMI spectra with a high spectral resolution solar irradiance (Caspar and Chance, 1997; Chance, 1998; Chance and Kurucz, 2010) using the preflight instrument slit functions (Dirksen *et al.*, 2006). Radiance wavelength calibration is performed for a representative swath line of radiance measurements (usually in the middle of the orbit) to determine a common wavelength grid for reference spectra.

Following wavelength correction, an undersampling correction spectrum is computed to partially correct for spectral undersampling (lack of Nyquist sampling: Chance, 1998; Slijkhuis *et al.*, 1999; Chance *et al.*, 2005). The calculation of the corrections for the undersampling is accomplished by convolving the preflight slit functions with the high-resolution solar spectrum and differencing its
fully-sampled and undersampled representations (Chance *et al.*, 2005).

To process each OMI orbit it is split into blocks of 100 swath lines. Spectral fitting is then performed for each block by processing the 60 cross-track pixels included in each swath line sequentially before advancing to the next swath line. The spectra are modeled as follows:

$$I = \left\{ (aI_0 + \sum_i \propto_i A_i)\, e^{-\sum_j (\beta_j B_j)} + \sum_k \gamma_k C_k \right\} Poly_{scale} + Poly_{baseline} \quad , \tag{1}$$

where $I_0$ is the solar irradiance (used in our operational BrO retrieval) or radiance reference measurement, $I$ is the Earthshine radiance (detected at satellite), $a$ is albedo, $\alpha_i$, $\beta_j$, $\gamma_k$, are the coefficients to the reference spectra of $A_i$, $B_j$, $C_k$, (for example, trace gas cross sections, Ring effect, vibrational Raman, undersampling correction, common mode, *etc.*) of model constituents. To

improve cross-track stripe biases (Section 3.4), the OMI daily solar irradiance ($I_0$) is substituted by the first principal component of the solar irradiances measured by OMI between 2005 and 2007 (one for each cross-track position). The principal component derived between 2005 and 2007 is used to process the entire mission. The reference spectra are derived separately for each cross-track position from original high-resolution cross sections convolved with the corresponding pre-

launch OMI slit functions (Dirksen *et al.*, 2006) after correcting for the solar $I_0$ effect (Aliwell *et al.*, 2002). Fig. 1 shows the trace gas cross sections and Ring spectra used in the current operational algorithm. The black lines are the original high-resolution reference spectra, and the color lines show the corresponding spectra convolved with OMI slit function, which are used in the fitting.

For improved numerical stability, radiances and irradiances are divided by their respective averages over the fitting window, renormalizing them to values of ~1. BrO is fitted in the spectral window 319.0–347.5 nm, within the UV-2 channel of the OMI instrument. The switch from the previous fitting window of 340–357.5 nm to this shorter and wider fitting window is based on extensive sensitivity analysis following the method described by Vogel et al., 2013. This new

fitting window aims at reducing the fitting uncertainty by including more BrO spectral structures as shown in Fig. 1 and reducing retrieval noise while preserving the stability of the algorithm. An analysis of the retrieval sensitivity to different windows is included in section 3.5.

The rotational Raman scattering (Chance and Spurr, 1997; Chance and Kurucz, 2010) and

undersampling correction spectra, $A_i$, are first added to the albedo-adjusted solar irradiance $aI_0$,

with coefficients $\alpha_i$ as shown in Eq. 1. Radiances $I$ are then modeled as this quantity attenuated by absorption from BrO, $O_3$, $NO_2$, $H_2CO$, and $SO_2$ with coefficients $\beta_j$ fitted to the reference spectra $B_j$ as shown in Eq. 1. A common mode spectrum $C_k$, computed on line, is added by fitting coefficient $\gamma_k$ after the Beer-Lambert law contribution terms. For each cross-track position, an

initial fit of all the pixels along the track between 30°N and 30°S is performed to determine the common mode spectra, derived as the average of the fitting residuals. The common mode spectra include any instrument effects that are uncorrelated to molecular scattering and absorption. This is done to reduce the fitting root-mean-square (RMS) residuals, and the overall uncertainties. These are then applied as reference spectra in fitting of the entire orbit. The fitting additionally contains

additive (*Poly_baseline*) and multiplicative closure polynomials (*Poly_scale*), parameters for spectral shift and, potentially, squeeze (not normally used). The operational parameters and the cross sections used are provided in Table 1.

   As part of the development of the OMBRO retrieval algorithm, a significant amount of effort was

dedicated to algorithm "tuning", i.e., the optimization of elements in the retrieval process, including interfering absorbers like $O_2$-$O_2$. The spectral region of 343 nm, where $O_2$-$O_2$ has an absorption feature larger than the BrO absorption, essentially is impossible to avoid in BrO retrievals: the fitting window would have to either terminate at shorter wavelengths or start past this feature, and both approaches yield to unacceptable low information content for the BrO

retrievals to succeed. During the tuning process, we investigated the effects of, among many other things, including or excluding $O_2$-$O_2$, the use of different spectroscopic data sets (Greenblatt et al., 1990 and Hermans et al., 1999 cross-sections), shorter or longer wavelength windows for the retrieval, and even extending the retrieval window beyond the $O_2$-$O_2$ absorption feature but excluding the approximate wavelength slice of the feature itself. The only approach that provided

quantitatively satisfactory results - i.e., stability of the retrieval under a wide range of conditions, minimized correlation with clouds, low fitting uncertainties, consistency of OMI global total column BrO with published results, and low noise in pixel-to-pixel retrievals - was to exclude $O_2$-$O_2$ from the OMBRO V3. It is difficult to quantify $O_2$-$O_2$ atmospheric content from the absorption feature around 343 nm alone, and its correlation with absorption bands of BrO and $H_2CO$ leads to

spectral correlations in the course of the non-linear least squares minimization process that are detrimental to the OMI BrO retrievals. Lampel *et al.*, (2018) provides spectrally resolved $O_2$-$O_2$

cross sections not only at 343 nm, but also at 328 nm (see Fig. 1) which is about 20% of the absorption at 343 nm and has not been shown in previous $O_2$-$O_2$ cross sections. Future updates to the operational OMBRO algorithm will investigate the effect of including Lampel *et al.*, (2018) $O_2$-$O_2$ cross sections on the fitting.

### 3.3 Air mass factors

Due to significant variation in $O_3$ absorption and Rayleigh scattering in the fitting window AMFs vary with wavelength by 10-15% as shown in Fig. 2. At large solar and viewing zenith angles it is difficult to identify a single representative AMF *ad hoc*. The wavelength dependent AMFs are

introduced to take into account for such strong variation within the BrO fitting window. They are applied pre-fit to the BrO cross sections, and the spectral fit retrieves VCDs directly. This direct fitting approach is a major departure from the commonly employed 2-step fitting procedure (OMI, 2002). It was first developed for retrievals of trace gases from SCIMACHY radiances in the shortwave infrared (Buchwitz *et al.*, 2000) and has been demonstrated for total $O_3$ and $SO_2$

retrievals from GOME/SCIAMACHY measurements in the ultraviolet (Bracher *et al.*, 2005; Coldewey-Egbers *et al.*, 2005; Weber et al., 2005; Lee *et al.,* 2008).

The albedo- and wavelength-dependent AMFs were pre-computed with the Linearized Discrete Ordinate Radiative Transfer code (LIDORT, Spurr, 2006) using a single mostly stratospheric BrO

profile (Fig. 3, left panel). The BrO profile, based on the model of Yung *et al.* (1980), has ~30% BrO below 15 km, ~10% BrO below 10 km, and ~2% BrO below 5 km. It should be noted that a fixed profile is inconsistent with the varying tropopause height (both with latitude and dynamically e.g. Salawitch *et al.* 2010) and therefore with the profile shape in the stratosphere, but the impact on the AMF is typically small as the scattering weight does not change much in the stratosphere.

For conditions with enhanced BrO in the lower troposphere, using this profile will overestimate the AMFs and therefore underestimate the BrO VCDs as discussed in Section 3.5. Surface albedos are based on a geographically varying monthly mean climatology derived from OMI (Kleipool *et al.,* 2008). Although AMFs based on this BrO profile only slightly depend on surface albedo, albedo effects can be significant over highly reflective snow/ice surfaces, reducing VCDs by 5-

30  10%.

In order to provide the AMF in the data product for consistency with previous versions based on a two-step approach, a second fitting of all OMI spectra is performed with unmodified BrO cross sections, which yields SCDs. An effective AMF can then be computed as AMF = SCD/VCD.

The green line in the top right panel of Fig. 1 shows the modified BrO cross section after multiplication with the wavelength-dependent AMF (albedo = 0.05, SZA (Solar Zenith Angle) = 5.0°, and VZA (Viewing Zenith Angle) = 2.5°). The wavelength-dependence in AMF is visible from the varying differences near BrO absorption peaks and the right wings at different

wavelengths. The correlation of the unmodified BrO cross sections with the rest of the fitted molecules is small (typically less than 0.12), except with $H_2CO$ (0.43). However, it is safe to assume that in most polar regions with enhanced BrO there are no high concentrations of formaldehyde. It will be worthwhile for future studies to assess the interference of $H_2CO$ under high $H_2CO$ and background BrO conditions similar to De Smedt et al., 2015. In addition, the AMF

wavelength dependence increases with the increase of solar and viewing zenith angles and surface albedo, which increases the correlation between modified BrO cross sections and $O_3$ cross sections. However, the correlation with $O_3$ becomes noticeable (~0.10) only at SZAs above ~80°.

### 3.4 Destriping

OMI L1b data exhibit small differences with cross-track position, due to differences in the dead/bad pixel masks (cross-track positions are mapped to physically separate areas on the CCD), dark current correction, and radiometric calibration, which lead to cross-track stripes in Level 2 product (Veihelmann and Kleipool, 2006). Our destriping algorithm employs several methods to reduce cross-track striping of the BrO columns. First, we screen outliers in the fitting residuals.

This method, originally developed to mitigate the effect of the South Atlantic Anomaly in SAO OMI BrO, $H_2CO$, and OClO data products, is now also being employed for GOME-2 (Richter *et al*., 2011). Screening outliers is done through computing the median, $r_{med}$, and the standard deviation $\sigma$ of residual spectra $r(\lambda)$ and in subsequent refitting excluding any spectral points for which $r(\lambda) \geq |r_{med} \pm 3\sigma|$. This can be done repeatedly for every ground pixel, which makes the

processing slow. However, we do it once for a reference swath line, recording the positions of the

bad pixels, and excluding them by default in each subsequent fit. Second, after the completion of the spectral fitting process for all ground pixels in the granule, a post-processing cross-track bias correction is performed: an average cross-track pattern is calculated from the along-track averages of all BrO VCDs for each cross-track position within a ±30° latitude band around the equator, to

which a low-order polynomial is fitted. The differences between the cross-track pattern and the fitted polynomial is then applied as a cross-track VCD correction (or "smoothing") factor. The smoothed VCDs are provided in a separate data field, *ColumnAmountDestriped*. Smoothed SCDs are derived in an analogous fashion and are also included in the data product.

**3.5 BrO VCD Error Analysis**

Estimated fitting uncertainties are given as $\sigma_i = \sqrt{C_{ii}}$ where $C$ is the covariance matrix of the standard errors. This definition is strictly true only when the errors are normally distributed. In the case where the level 1 data product uncertainties are not reliable estimates of the actual uncertainties, spectral data are given unity weight over the fitting window, and the $1\sigma$ fitting error

in parameter $i$ is determined as

$$\sigma_i = \varepsilon_{rms}\sqrt{\frac{c_{ii} \times npoints}{npoints - nvaried}} \tag{2}$$

where $\varepsilon_{rms}$ is the root mean square of the fitting residuals, *npoints* is the number of points in the fitting window, and *nvaried* is the number of parameters varied during the fitting.

The fitting uncertainties for single measurements of the BrO VCDs typically vary between $4\times10^{12}$ and $7\times10^{12}$ molecules cm$^{-2}$, consistently throughout the data record. The uncertainties vary with cross-track positions, from ~$7\times10^{12}$ at nadir positions to ~$4\times10^{12}$ at edge positions due to the increase of photon path length through the stratosphere. Relatively, the VCD uncertainties typically range between 10-20% of individual BrO VCDs, but could be as low as 5% over BrO

hotspots. This is roughly 2-3 times worse that what was achieved from GOME-1 data.

The BrO VCD retrieval uncertainties listed in the data product only include random spectral fitting errors. Error sources from AMFs (*i.e.*, BrO climatology), atmospheric composition and state

(pressure/temperature vertical profiles, total $O_3$ column, *etc.*) and other sources of VCD uncertainty are not included. We provide here error estimates for these additional error sources.

Uncertainties in the AMFs, used to convert slant to vertical columns, are estimated to be 10% or
less except when there is substantially enhanced tropospheric BrO. Hence the total uncertainties of the BrO vertical columns typically range within 15-30%. To estimate the AMF error associated with enhanced tropospheric concentrations we have studied the difference between AMFs calculated using the stratospheric only BrO profile and a stratospheric-tropospheric profile as shown in the right panel of Fig. 3. Fig. 4 shows the dependency of the relative AMF difference
with respect to wavelength (top panel), albedo (middle panel) and VZA (bottom panel) as a function of the SZA between calculations performed using these two profiles. The use of stratospheric only BrO profile can lead to AMF errors up to 50% depending on albedo and viewing geometry. On average, using the stratospheric only BrO profile overestimates AMF and underestimates VCD by 41%.

We have performed sensitivity analysis of OMI BrO VCD with respect to various retrieval settings using orbit 26564 on 13 July 2009. Table 2 shows the median VCDs, median fitting uncertainties and the number of negative VCD pixels for each configuration. Table 3 summarizes the overall fitting error budget including the random fitting uncertainty, cross sections errors (as reported in
the literature), and various retrieval settings. We studied five wavelength windows including the current operational window (319.0-347.5 nm) version 2 window (323.0-353.5 nm), version 1 (340.0-357.5 nm) and two extra windows exploring the impact of extending the window to shorter wavelengths (310.0-357.5 nm) and reducing it by limiting its extension to wavelengths above 325 nm (325.0-357.5). The choice of fitting window can cause significant differences in BrO VCDs of
up to 50%. The current window results in the most stable retrievals with the smallest number of pixels with negative VCD values.

Including the interference of $O_2$-$O_2$ leads to a decrease of the median VCD by ~12% and an increase of the median fitting uncertainty by ~10% with respect to the operational set up. Excluding
$H_2CO$ from the fitting significantly reduces the retrieved BrO columns by ~37%, given that the strong anticorrelation between both molecules is not taken into account. Fitting the mean residual

(common mode) has a small impact in the retrieval results, the median VCD only changes ~3%, but reduces the median fitting uncertainty by ~30% with respect to the exclusion of the common mode. To study the impact of the slit functions we have performed the retrieval using both online slit functions, modelled as a Gaussian, and the preflight instrument slit functions. The median

difference between these two retrievals is 27% for orbit number 26564. We have investigated the impacts of the order of scaling and baseline polynomials; it can cause uncertainties of ~10% as shown in Table 3.

To study the impact of the radiative transfer effects of the $O_3$ absorption in our retrieval we have

adopted the correction method described by Pukite et al., 2010. We find that between 60° south and 60° north the average difference is smaller than 10% with values around 2% near the equator. However, as we move near the poles with solar zenith angles above 60° the differences start to be bigger arriving to mean values around 30%.

## 4 Results and discussions

Comparisons of the OMI OMBRO product with GOME-2 satellite retrievals and remote sensing ground based measurements over Harestua, Norway as well as monthly mean averages illustrate the quality of the retrieval on a global scale. On a local scale, recent scientific studies looking at BrO enhancements in volcanic plumes and over salt lakes are pushing the limits of the current OMBRO setups. In the following sections, we provide details of these comparisons (section 4.1)

and discuss OMI OMBRO global distribution (section 4.2) and local enhancements over salt lakes and volcanic plumes observations (section 4.3), and their applicability and strategies to correctly use the publicly available OMBRO product.

### 4.1 Comparisons with GOME-2 and ground-based observations

To assess the quality of the OMBRO product, we first compared OMI BrO VCDs with

BIRA/GOME-2 BrO observations (Theys *et al.*, 2011). GOME-2 has descending orbit with a local equator crossing time (ECT) of 9:30 am and OMI has ascending orbit with an ECT of 1:45 pm. To minimize the effects of diurnal variation especially under high SZAs (e.g., McLinden *et al.*, 2006; Sioris *et al.*, 2006) on the comparison, we conduct the comparison using simultaneous nadir

overpasses (SNOs) within 2 minutes between GOME-2 and OMI predicted by NOAA National Calibration Center's SNO prediction tool (https://ncc.nesdis.noaa.gov/SNOPredictions). Given Aura and Metop-A satellite orbits, all these SNOs occur at high latitudes around 75°S/N. Fig. 5 shows the time series of comparison of individual OMI/GOME-2 BrO retrievals from February
2007 through November 2008. The temporal variation of BrO at the SNO locations is captured similarly by OMI and GOME-2 BrO. The scatter plot in Fig. 6 quantifies the comparison between OMI and GOME-2 BrO. OMI BrO shows excellent agreement with GOME-2 BrO with a correlation of 0.74, and a mean bias of -0.216 ± 1.13×10$^{13}$ molecules cm$^{-2}$ (mean relative bias of -2.6 ± 22.1%). Considering very different retrieval algorithms including different cross sections
and BrO profiles, such a good agreement is remarkable. GOME-2 retrievals use the BrO cross sections of Fleischmann *et al.* (2004) while our BrO retrievals use the BrO cross sections of Wilmouth *et al.* (1999). According to the sensitivity studies by Hendrick *et al.* (2009), using the Fleischmann cross section increases BrO by ~10%. So, accounting for different cross sections, OMI BrO underestimates the GOME-2 BrO by ~10%. In addition, the GOME-2 algorithm uses a
residual technique to estimate tropospheric BrO from measured BrO SCDs by subtracting a dynamic estimate of stratospheric BrO climatology driven by $O_3$ and $NO_2$ concentrations and by using two different tropospheric BrO profiles depending on surface albedo conditions. This is very different from the approach of using a single BrO profile in the OMI BrO algorithm, and can contribute to some of the BrO differences. Furthermore, additional algorithm uncertainties in both
algorithms and different spatial sampling can also cause some differences. Fig. 7 shows the VCDs monthly averages of GOME-2 data (green) and OMBRO (black) from February 2007 to December 2009 where the seasonal variations are clearly seen. Our study shows that OMI has negative mean biases of 0.35×10$^{13}$ molecules cm$^{-2}$ (12%), 0.33×10$^{13}$ molecules cm$^{-2}$ (10%), 0.25×10$^{13}$ molecules cm$^{-2}$ (17%), and 0.30×10$^{13}$ molecules cm$^{-2}$ (10%) for Alaska, Southern Pacific, Hudson Bay, and
Greenland, respectively.

We also used ground-based zenith-sky measurements of total column BrO at Harestua, Norway (Hendrick *et al.*, 2007) to estimate the quality of the OMI BrO. We compared daily mean total BrO at Harestua with the mean OMI BrO from individual footprints that contain the location of
Harestua site. Fig. 8 shows the time series of the comparison between OMI total BrO and Harestua total BrO from February 2005 through August 2011 with the scatter plot shown in Fig. 9. Ground-

based BrO shows an obvious seasonality with high values in the winter/spring and low values in the summer/fall. Such seasonality is well captured by OMI BrO. OMI BrO shows a reasonable good agreement with Harestua BrO with a correlation of 0.46 and a mean bias of $0.12\pm0.76\times10^{13}$ molecules cm$^{-2}$ (mean relative bias of $3.18\pm16.30\%$, with respect to individual Harestua BrO).

Sihler *et al.* (2012) compared GOME-2 BrO to ground-based observations at Utqiagvik (Barrow) finding the correlation to be weaker (r = 0.3), likely due to both elevated and shallow surface layers of BrO. However, their correlation between GOME-2 BrO and ground-based measurements made from the Icebreaker Amundsen, in the Canadian Arctic Ocean  (r = 0.4) is closer to our correlation here. From the Harestua data, tropospheric BrO typically consists of 15-30% of the total BrO,

larger than what we have assumed in the troposphere. The use of a single BrO profile in the OMI BrO algorithm will likely underestimate the actual BrO. Accounting for the uncertainty due to profile shape, OMI BrO will have a larger positive bias relative to Harestua measurements, which can be caused by other algorithm uncertainties and the spatiotemporal differences between OMI and Harestua BrO.

## 4.2 Global distribution of BrO VCDs

Fig. 10 presents the global distribution of monthly mean BrO VCDs for selected months (March, June, September, December) showing BrO seasonality for three different years (2006, 2007 and 2012). BrO typically increases with latitude, with minimal values in the tropics (~$2\times10^{13}$ molecules

cm$^{-2}$) and maximum values (~$10^{14}$ molecules cm$^{-2}$) around polar regions especially in the northern hemisphere winter/spring. In the tropics, BrO shows little seasonality but at higher latitudes in polar regions, BrO displays evident seasonality. The seasonality is different between northern and southern hemispheres. In the northern hemisphere, BrO values are larger in spring (March) with widespread enhancement and smaller in fall (September/December). In the southern hemisphere,

BrO values are larger in southern hemispheric spring and summer (i.e., September and December) and smaller in the winter (June). Such global distribution and seasonal variation are generally consistent with previous satellite measurements (*cf.* Chance, 1998; http://bro.aeronomie.be/level3_monthly.php?cmd=map). BrO in the tropics shows consistent zonal distributions with lower values over land and in the intertropical convergence zone.  This

might be related to the impacts of clouds on the retrievals (e.g, BrO below thick clouds cannot be

measured, there are uncertainties in the AMF calculation under cloudy conditions) and will be investigated in detail in future studies. The global distribution and seasonal variation are consistent from year to year, but the distributions from different years disclose some interannual variation. For example, BrO values in 2007 are smaller in January but are larger in March compared to those

in 2006. Although OMI data since 2009 have been seriously affected by the row anomaly at certain cross-track positions, the monthly mean data derived from good cross-track positions are hardly affected by the row anomaly as shown from the very similar global distribution and seasonality in 2012.

**4.3 Salt lakes and volcanic plumes enhancements of BrO**

Following recent work by Hörmann *et al*. (2016) over the Rann of Kutch using OMI BrO retrievals from an independent research product we have explored the capability of our OMBRO product to observe similar enhancements in other salt lakes. Fig. 11 shows monthly averaged OMI BrO over the Great Salt Lake for 06/2006, the corresponding surface albedo used in the retrieval, cloud cover (assuming a cloud filter of 40%) as well as the cloud pressure. Over the Great Salt Lake, BrO

enhancement occurs predominantly over the lake bed with enhancements of $\sim$5-10$\times$10$^{12}$ molecules cm$^{-2}$ over background values (3-4$\times$10$^{13}$ molecules cm$^{-2}$). Despite observing these enhancements, the users of OMBRO for these kinds of studies should be aware of three limitations of the current retrieval algorithm. First, the BrO columns assume a mostly stratospheric BrO profile (Fig. 3 left panel) for the AMF calculation. Second, the OMI derived albedo climatology (Kleipool *et al*.

2008) used in OMBRO has a resolution of 0.5°. At this resolution OMBRO retrievals can have biases given the size of OMI pixels and the inherent sub-pixel albedo variability. Finally, high albedos inherent to salt lakes surface yield abnormally high cloud fractions and low cloud pressures over the salt lakes (Hörmann *et al*., 2016). All these factors should be considered in studies addressing the spatiotemporal distribution of BrO over salt lakes using OMBRO.

During our analysis of volcanic eruption scenarios, it was discovered that the currently implemented $SO_2$ molecular absorption cross sections (Vandaele *et al*., 1994) are a sub-optimum choice (see Fig. 12). Compared to more recent laboratory measurements (Hermans *et al*., 2009; Vandaele *et al*., 2009), the original $SO_2$ cross sections implemented in OMBRO do not extend

over the full BrO fitting window and exhibit the wrong behavior longward of 324 nm,

overestimating the most recent measurement by up to a factor of 3. As the correlation between BrO and both $SO_2$ cross sections are very small (-0.03 for the current $SO_2$ and 0.11 for the latest $SO_2$ cross sections) over the spectral range of $SO_2$ cross sections, interference by $SO_2$ in BrO retrievals is usually not an issue at average atmospheric $SO_2$ concentrations, but strong volcanic

eruptions will render even small $SO_2$ absorption features past 333 nm significant. Around 334 nm, the Vandaele *et al*. (2009) data show an $SO_2$ feature that correlates with BrO absorption when $SO_2$ concentrations are significantly enhanced. As a consequence of this spectral correlation, $SO_2$ may be partially aliased as BrO, since the implemented $SO_2$ cross sections cannot account for it. Fig. 13 presents an example from the 2010 Eyjafjallajökull eruption to show that the BrO retrieval can

be affected by the choice of $SO_2$ cross sections. The next version of the OMBRO public release will be produced using the updated $SO_2$ absorption cross sections. Until then, caution is advised when using the OMI BrO product during elevated $SO_2$ conditions. We recommend to use OMBRO product together with the operational OMI $SO_2$ product (Li *et al.*, 2013) to flag abnormally high BrO retrievals.

The top panels of Fig. 13 show daily average operational BrO VCDs from the eruption of the Eyjafjallajökull volcano on May 5 and 17, 2010, respectively. Enhanced BrO values in excess of $8.0 \times 10^{13}$ molecules $cm^{-2}$ are detected in the vicinity of this volcano (e.g., plume extending southeast ward from the volcano on May 5 and, high BrO over Iceland on May 17). Some of these

enhanced BrO values correspond to the locations of enhanced $SO_2$ as shown from the NASA global $SO_2$ monitoring website (https://so2.gsfc.nasa.gov/). This enhancement of BrO is not related to the seasonal variation of BrO as no such BrO enhancement is detected over Eyjafjallajökull during May 5-17, 2011 (a year after the eruption), with BrO values of only up to ~$5.3 \times 10^{13}$ molecules $cm^{-2}$ (not shown). The bottom panels of Fig. 13 show the same BrO retrievals using $SO_2$ cross sections

by Vandaele et al. (2009). Using the improved $SO_2$ cross sections increase the BrO over a broader area on both days, supporting that the choice of $SO_2$ cross sections can affect the BrO retrievals. However, BrO enhancement around the volcano can still clearly be seen with the improved $SO_2$ cross sections. This suggests that this BrO enhancement is not totally due to aliasing of $SO_2$ as BrO, but potentially real BrO from the volcanic eruption.

## 5 Conclusions

This paper describes the current operational OMI BrO retrieval algorithm developed at SAO and the corresponding V3 OMI total BrO (OMBRO) product in detail. The OMI BrO retrieval algorithm is based on nonlinear least-squares direct fitting of radiance spectra in the spectral range

319.0-347.5 nm to obtain vertical column densities (VCDs) directly in one step. Compared to previous versions of two-step algorithms, the fitting window was moved to shorter wavelengths and the spectral range was increased to reduce the fitting uncertainty. Because air mass factors (AMFs) vary significantly with wavelengths as a result of significant variation of $O_3$ absorption, the wavelength and surface albedo dependent AMF, which is precomputed with the Linearized

Discrete Ordinate Radiative Transfer (LIDORT) code using a single mostly stratospheric BrO profile, is applied pre-fit to BrO cross sections for direct fitting of VCDs. Prior to the spectral fitting of BrO, wavelength calibration is performed for both irradiance and radiance at each cross-track position and reference spectra are properly prepared at the radiance wavelength grid. Then radiances are modeled from the measured solar irradiance, accounting for rotational Raman

scattering, undersampling, attenuation from BrO and interfering gases, and including additive and multiplicative closure polynomials, and the average fitting residual spectrum. To maintain consistency with previous versions, a second fitting of all OMI spectra is performed with unmodified BrO cross sections to derive SCDs and the effective AMFs. Then a destriping step is employed to reduce the cross-track dependent stripes.

The uncertainties of BrO VCDs included in the data product include only spectral fitting uncertainties, which typically vary between 4 and $7\times10^{12}$ molecules cm$^{-2}$ (10-20% of BrO VCDs, could be as low as 5% over BrO hotspots), consistent throughout the data record. The uncertainties vary with cross-track positions, from $\sim7\times10^{12}$ at nadir positions to $\sim4\times10^{12}$ at edge positions. We

have investigated additional fitting uncertainties caused by interferences from $O_2$-$O_2$, $H_2CO$, $O_3$, and $SO_2$, the impact of the choice of fitting window, the use of common mode, the orders of closure polynomials, and instrument slit functions. Uncertainties in the AMF calculations are estimated at $\sim10\%$ unless the observation is made over a region with high tropospheric BrO columns. In this case, the use of a single stratospheric BrO profile is another source of uncertainty, overestimating

AMFs (up to 50%) and therefore underestimating BrO VCDs.

We compared OMI BrO VCDs with BIRA/GOME-2 BrO observations at locations of simultaneous nadir overpasses (SNOs), which only occur around 75ºN and 75ºS. OMI BrO shows excellent agreement with GOME-2 BrO with a correlation of 0.74, and a mean bias of -$0.216\pm1.13\times10^{13}$ molecules $cm^{-2}$ (mean relative bias of -2.6 ± 22.1%). Monthly mean OMBRO

VCDs during 2007-2009 show negative biases of $0.25\text{-}0.35\times10^{13}$ molecules $cm^{-2}$ (10-17%) over Alaska, Southern Pacific, Hudson Bay, and Greenland, respectively. We also compared OMI BrO with ground-based zenith-sky measurements of total BrO at Harestua, Norway. The BrO seasonality in Harestua is well captured by the OMI BrO and OMBRO retrieval showing a reasonable good agreement with the ground-based measurements. The correlation between both

datasets is of 0.46 and the mean bias $0.12\pm0.76\times10^{13}$ molecules $cm^{-2}$ (mean relative bias of 3.18±16.30%).

The global distribution and seasonal variation of OMBRO are generally consistent with previous satellite measurements. There are small values in the tropics with little seasonality, and large

values at high latitudes with distinct seasonality. The seasonality is different between the northern and southern hemisphere, with larger values in the hemispheric winter/spring (spring/summer) and smaller values in summer/fall (winter) for the northern (southern) hemisphere. This spatiotemporal variation is generally consistent from year to year and is hardly affected by the row anomaly, but does show some interannual variation. Finally, we have explored the feasibility of detecting

enhanced BrO column over salt lakes and in volcanic plumes using OMBRO retrievals. We found enhancement of the BrO with respect to the background levels of $5\text{-}10\times10^{12}$ molecules $cm^{-2}$ over the U.S. Great Salt Lake. We also observed a significant enhancement from the eruption of Eyjafjallajökull volcano although BrO retrievals under high $SO_2$ conditions can be affected by the current use of a sub-optimal choice of $SO_2$ cross sections.

Several important retrieval issues in the current operational algorithm that affect the quantitative use BrO VCDs have been raised in this paper such as the exclusion of $O_2\text{-}O_2$, nonoptimal $SO_2$ cross sections, the neglect of radiative effect of $O_3$ absorption, and the assumption of stratospheric only BrO profile. The users are advised to pay attention to these issues so that the product can be

used properly. Future versions of OMBRO will include updated $SO_2$ and $O_2\text{-}O_2$ cross sections, corrections for the radiative transfer effect of the $O_3$ absorption and reoptimize the spectral fitting

windows to mitigate the interferences of other trace gases. We will also improve the AMF calculation accounting for clouds and $O_3$ and will consider the use of model-based climatological BrO profiles. These updates will increase the capabilities of the OMBRO retrieval to quantitatively estimate enhancements over salt lakes and in volcanic plumes.

**Acknowledgements**

This study is supported by NASA Atmospheric Composition Program/Aura Science Team (NNX11AE58G) and the Smithsonian Institution. Part of the research was carried out at the Jet Propulsion Laboratory, California Institute of Technology, under a contract with NASA. The Dutch-Finnish OMI instrument is part of the NASA EOS Aura satellite payload. The OMI project is managed by NIVR and KNMI in the Netherlands. We acknowledge the OMI International Science Team for providing the SAO OMBRO data product used in this study.

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

**Table 1. Fitting window and parameters used to derive BrO vertical column densities**

| Parameter | Description/value |
|---|---|
| Fitting window | 319.0 - 347.5 nm |
| Baseline polynomial | 4th order |
| Scaling polynomial | 4th order |
| Instrument slit function | Hyper-parameterization of pre-flight measurements, Dirksen *et al*., 2006 |
| Wavelength calibration | Spectral shift (no squeeze) |
| Solar reference spectrum | Chance and Kurucz, 2010 |
| BrO cross sections | Wilmouth *et al*., 1999, 228K |
| $H_2CO$ cross sections | Chance and Orphal, 2011, 300K |
| $O_3$ cross sections | Malicet *et al*., 1995, 218K, 295K |
| $NO_2$ cross sections | Vandaele *et al*., 1998, 220K |
| $SO_2$ cross sections | Vandaele et al., 1994, 295K[1] |
|  | Hermans *et al*., 2009; Vandaele *et al*., 2009, 295K[2] |
| OClO cross sections | Kromminga *et al*., 2003, 213K |
| Molecular Ring cross sections | Chance and Spurr, 1997 |
| Undersampling correction | Computed on-line, Chance *et al*., 2005 |
| Residual (common mode) spectrum | Computed on-line between 30°N and 30°S |

1. Used in the current operational algorithm.

2. Used for testing sensitivity to $SO_2$ cross sections and will be used in the next version.

**Table 2. Error analysis studies. For reference, the total number of retrieved pixels not affected by the row anomaly is 58112.**

| Description | Median VCD (Molec. cm$^{-2}$) | Median uncertainty (Molec. cm$^{-2}$) | Number of negatives |
|---|---|---|---|
| 319 - 347.5 nm Op. (V3) | $4.02\times10^{13}$ | $7.11\times10^{12}$ | 88 |
| 323.0 - 353.5 nm (V2) | $2.65\times10^{13}$ | $9.27\times10^{12}$ | 1604 |
| 340.0 - 357.5 nm (V1) | $2.86\times10^{13}$ | $1.19\times10^{13}$ | 3351 |
| 310.0 - 357.5 nm | $1.97\times10^{13}$ | $6.18\times10^{12}$ | 2728 |
| 325.0 - 357.5 nm | $3.16\times10^{13}$ | $8.02\times10^{12}$ | 1416 |
| With $O_2$-$O_2$ | $3.54\times10^{13}$ | $7.80\times10^{12}$ | 319 |
| Online slit function | $5.09\times10^{13}$ | $7.16\times10^{12}$ | 68 |
| Without common mode | $3.89\times10^{13}$ | $1.02\times10^{13}$ | 116 |
| Without $H_2CO$ | $2.52\times10^{13}$ | $6.27\times10^{12}$ | 816 |

**Table 3. Summary of different errors sources in the BrO vertical column.**

| Error source | Type | Parameter uncertainty | Averaged uncertainty on BrO VCD | Evaluation method - reference |
|---|---|---|---|---|
| Measurement noise random | Random | S/N 500 - 1000 | $4\text{-}7\mathrm{x}10^{12}$ molec. cm$^{-2}$ | Error propagation; |
| H$_2$CO | Systematic | Based on literature reported error estimates | 5% | Chance and Orphal, 2011, 300K |
| O$_3$ | | | 2% | Malicet et al., 1995, 218K, 295K |
| BrO | | | 8% | Wilmouth et al., 1999, 228K |
| NO$_2$ | | | 3% | Vandaele et al., 1998, 220K |
| SO$_2$ | | | 5% | Vandaele et al., 1994, 295K |
| OClO | | | 5% | Kromminga et al., 2003, 213K |
| Ring | | | 5% | Chance and Spurr, 1997 |
| Order of baseline polynomial | Systematic | Vary polynomial order | 10% | Sensitivity analysis |
| Order of scaling polynomial | Systematic | Vary polynomial order | 10% | |
| Instrumental slit function and wavelength calibration | Systematic | Preflight and online slit function | 27% | |
| Wavelength interval | Systematic | Varying fitting window | 50% | |

**Figures and Figure Captions**

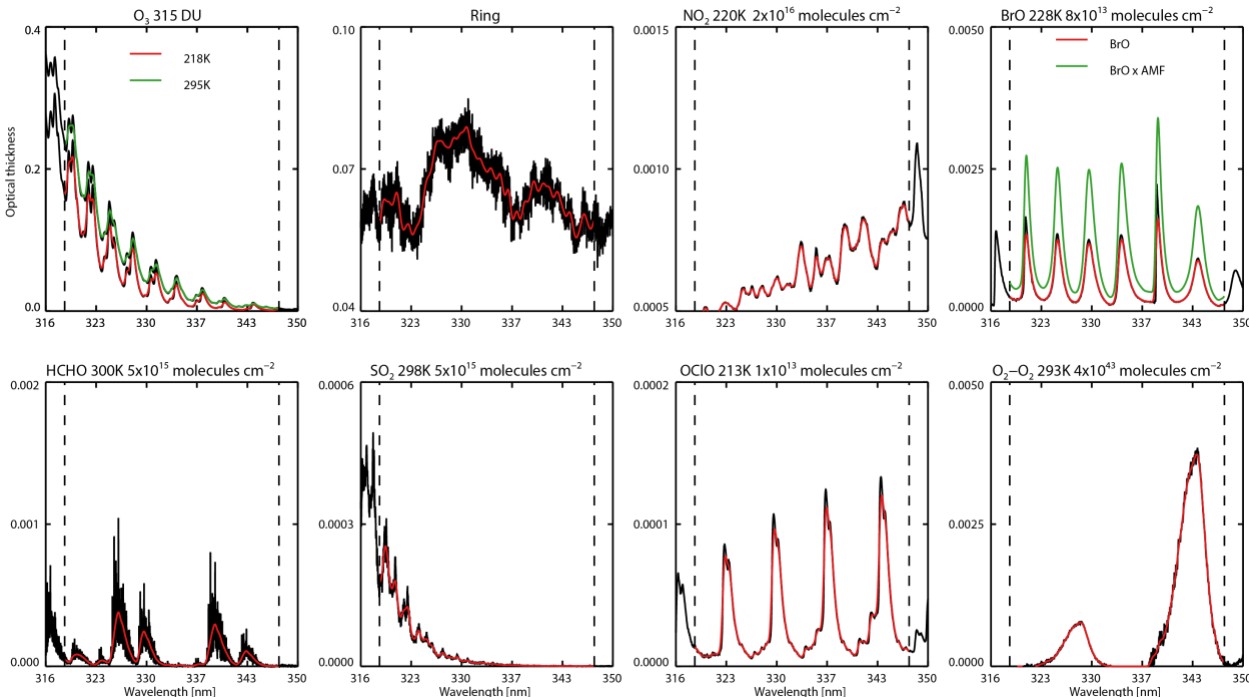

Figure 1. Cross sections used in the current operational BrO algorithm except for the SO₂ cross section at 298 K which is to be used in the next version. The black lines are the original cross sections, the color lines show the cross sections convolved with approximate OMI slit function (which is assumed to be a Gaussian with 0.42nm full width at half maximum). The O₂-O₂ calculation is based on Lampel *et al.* (2018) cross sections. The BrO cross section after multiplication with the wavelength-dependent AMFs used these parameters for the AMF calculation: albedo = 0.05, SZA= 5.0º, and VZA = 2.5º). The RMS of the fitting residuals are on the order of 9x10⁻⁴, indicating that BrO spectral features are slighly bigger than typical fitting residuals.

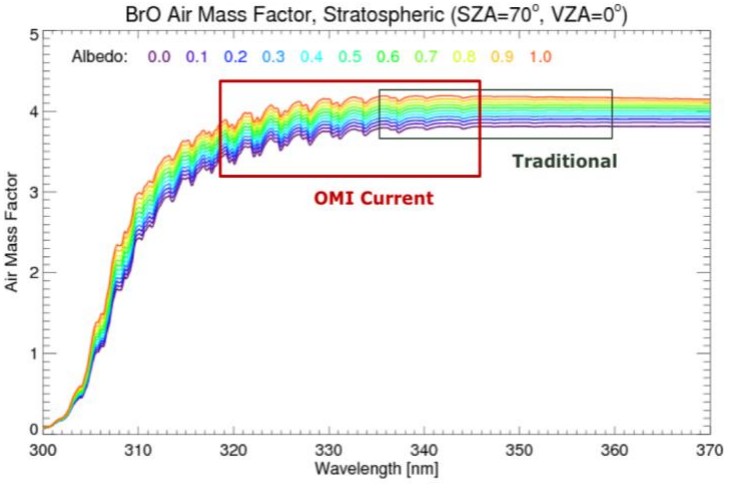

**Figure 2. Wavelength- and albedo-dependent air mass factors calculated using a mostly stratospheric fixed BrO profile. The blue box shows the fitting window used in our previous versions, and the red box shows the fitting window used in the current operational algorithm.**

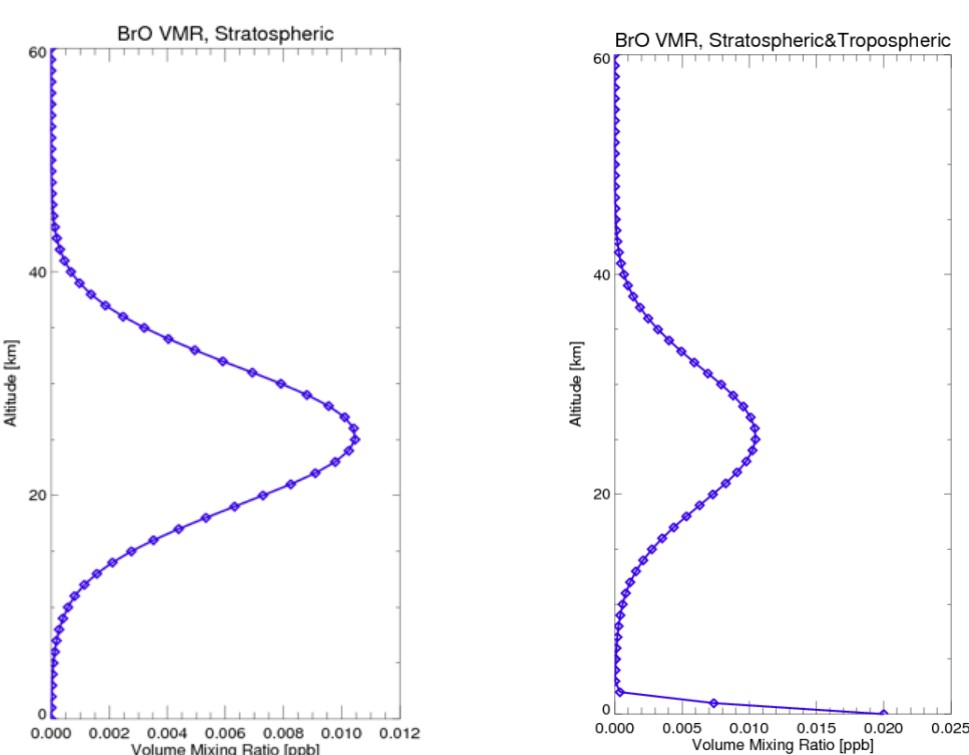

**Figure 3. (Left) A mostly stratospheric vertical BrO profile used for air mass factors calculations in OMBRO V3. Total BrO, BrO < 15 km, BrO < 10 km, and BrO < 5km are $2.05 \times 10^{13}$, $5.06 \times 10^{12}$, $1.55 \times 10^{12}$, and $2.87 \times 10^{11}$ molecules cm$^{-2}$, respectively. (Right) A stratospheric tropospheric vertical BrO profile used to investigate the impact of high tropospheric BrO columns on air mass factors calculations. Total BrO, BrO < 15 km, BrO < 10 km, and BrO < 5km are $6.99 \times 10^{13}$, $5.45 \times 10^{13}$, $5.10 \times 10^{13}$, and $4.97 \times 10^{13}$ molecules cm$^{-2}$, respectively.**

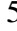

**Figure 4. The percentage of relative AMF errors as a function of the SZA and the wavelength (top panel), albedo (middle panel) and VZA (bottom panel) when using the stratospheric only BrO profile (Fig. 3, left panel) in the case there exists a significant tropospheric BrO**
10   **column as shown in the stratospheric-tropospheric BrO profile (Fig. 3, right panel).**

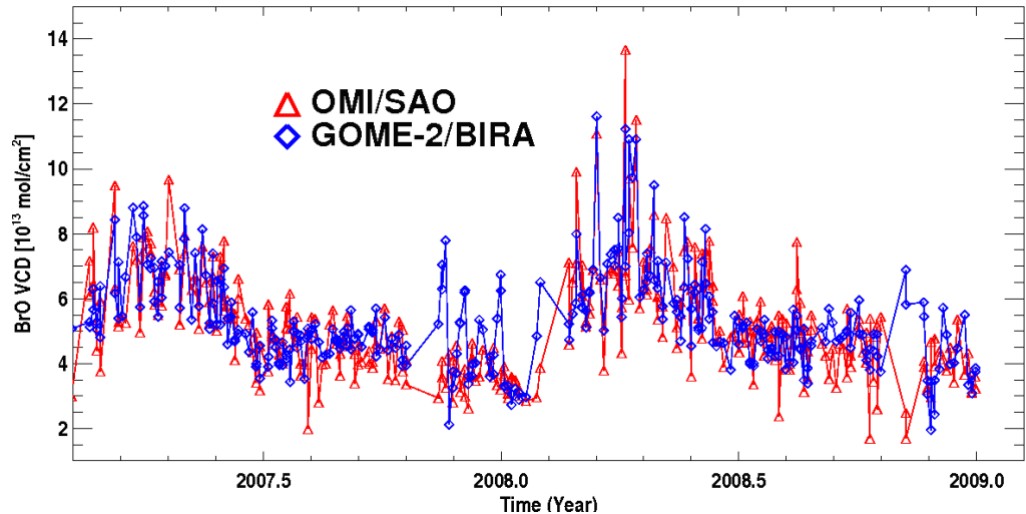

**Figure 5. Time series comparison of SAO OMI (red) BrO and BIRA GOME-2 (blue) BrO VCDs from February 2007 to November 2008 using simultaneous nadir overpasses occurring at high latitudes, around 75° S/N, and within 2 minutes between OMI and GOME-2 observations.**

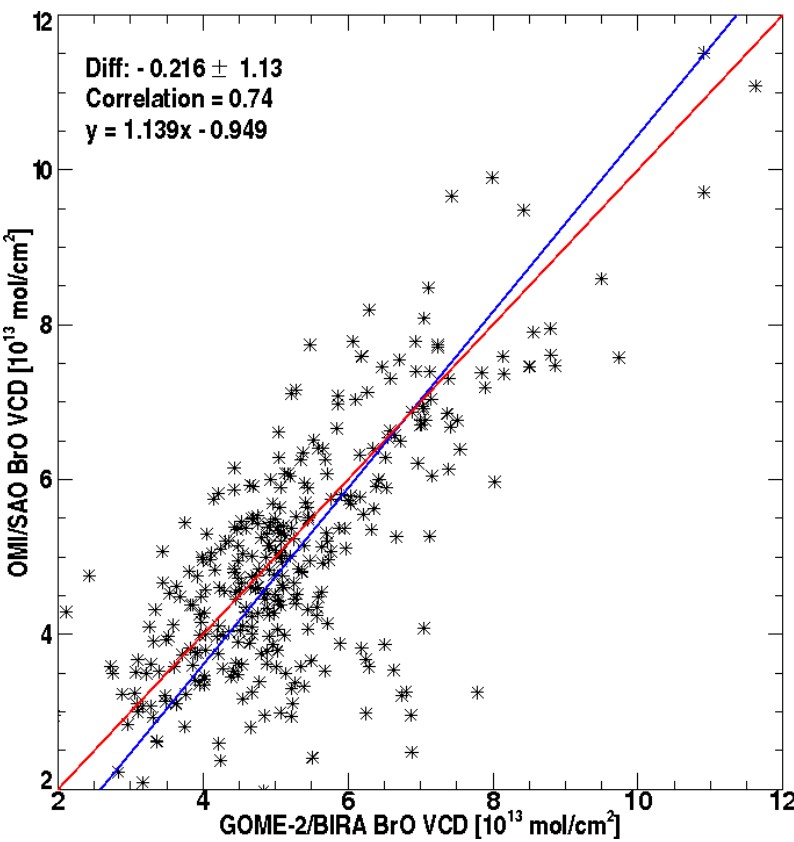

**Figure 6. Correlation and orthogonal regression of OMI and GOME-2 BrO for the data shown in Fig. 5. The legends show the mean bias and standard deviation of the differences, correlation, and the orthogonal regression.**

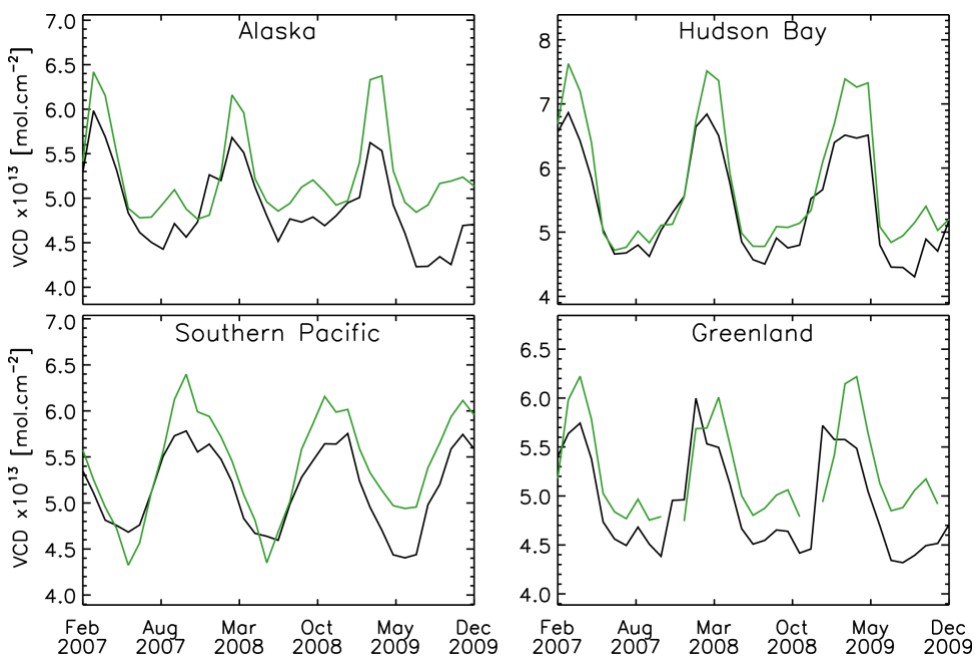

**Figure 7. VCD of GOME-2 (green) comparison to OMI (black) over four regions from February 2007 to December 2009 for four regions. Each region is defined by a square with the following latitude/longitude boundaries: Alaska (50ºN-70ºN/165ºW-135ºW), Hudson Bay (50ºN-65ºN/95ºW-75ºW), Southern Pacific (70ºS-50ºS/135ºE-155ºE), and Greenland (60ºN-80ºN/60ºW-15ºW).**

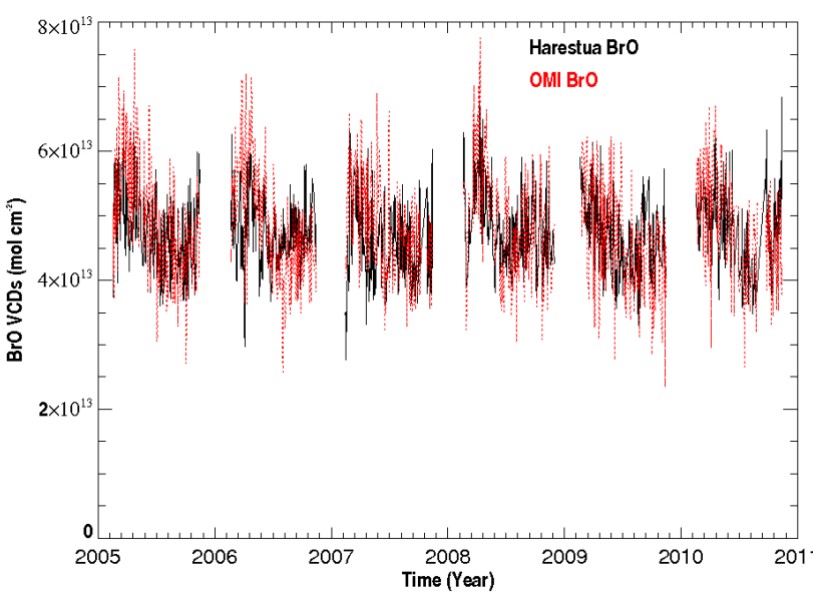

**Figure 8. Time series comparison of ground-based zenith-sky total BrO (black) at Harestua, Norway and coincident SAO OMI BrO (red) from February 2005 through August 2011.**

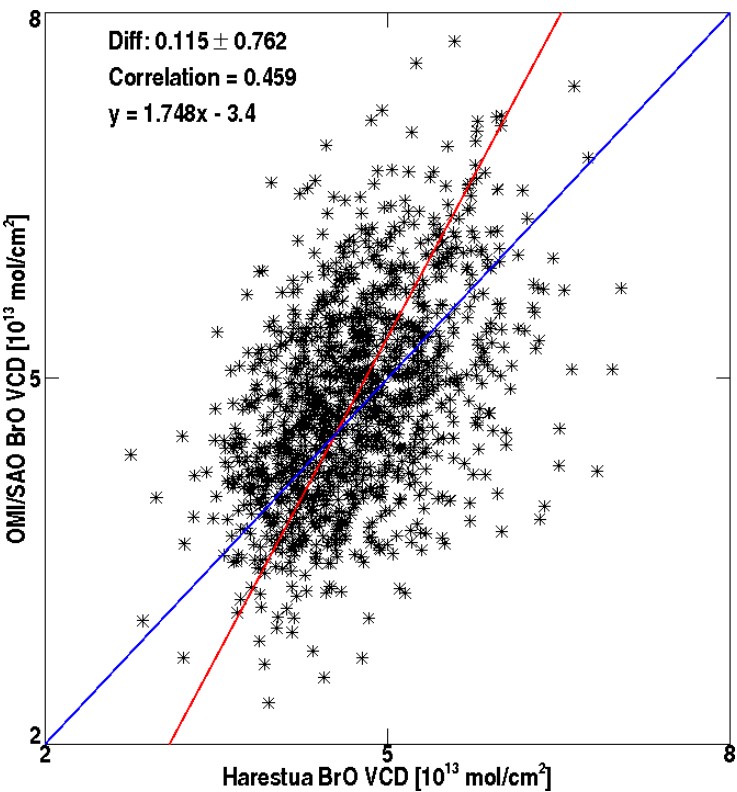

**Figure 9. Correlation and orthogonal regression of OMI and Harestua BrO for the data in Fig. 8. The legends show the mean biases and standard deviations of the differences, correlation, and the orthogonal regression.**

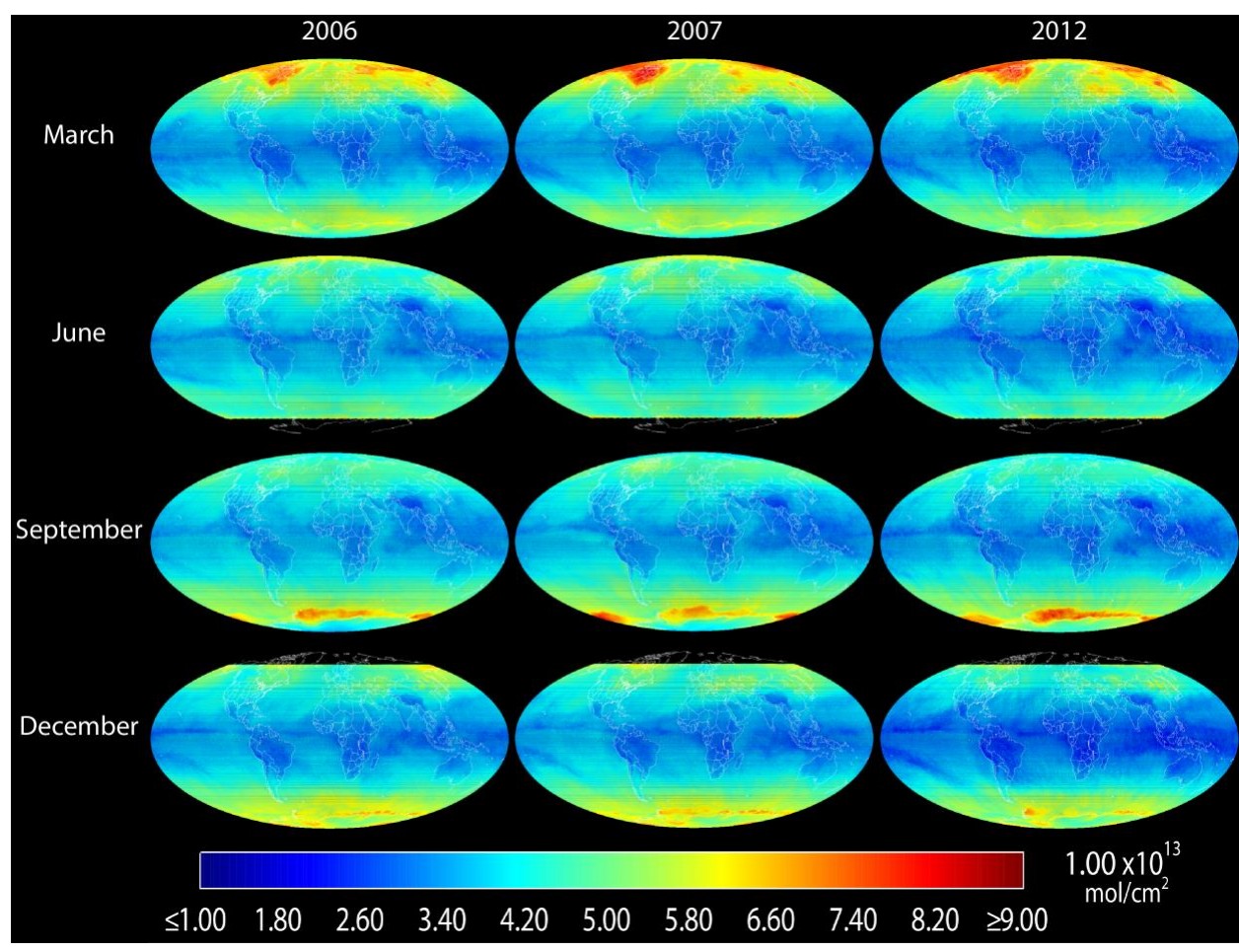

**Figure 10. Global distributions of monthly mean BrO VCDs in March, June, September and December (in different rows) of 2006, 2007, and 2012 (different columns). Bromine release "explosions" during the Polar Spring months can be seen clearly.**

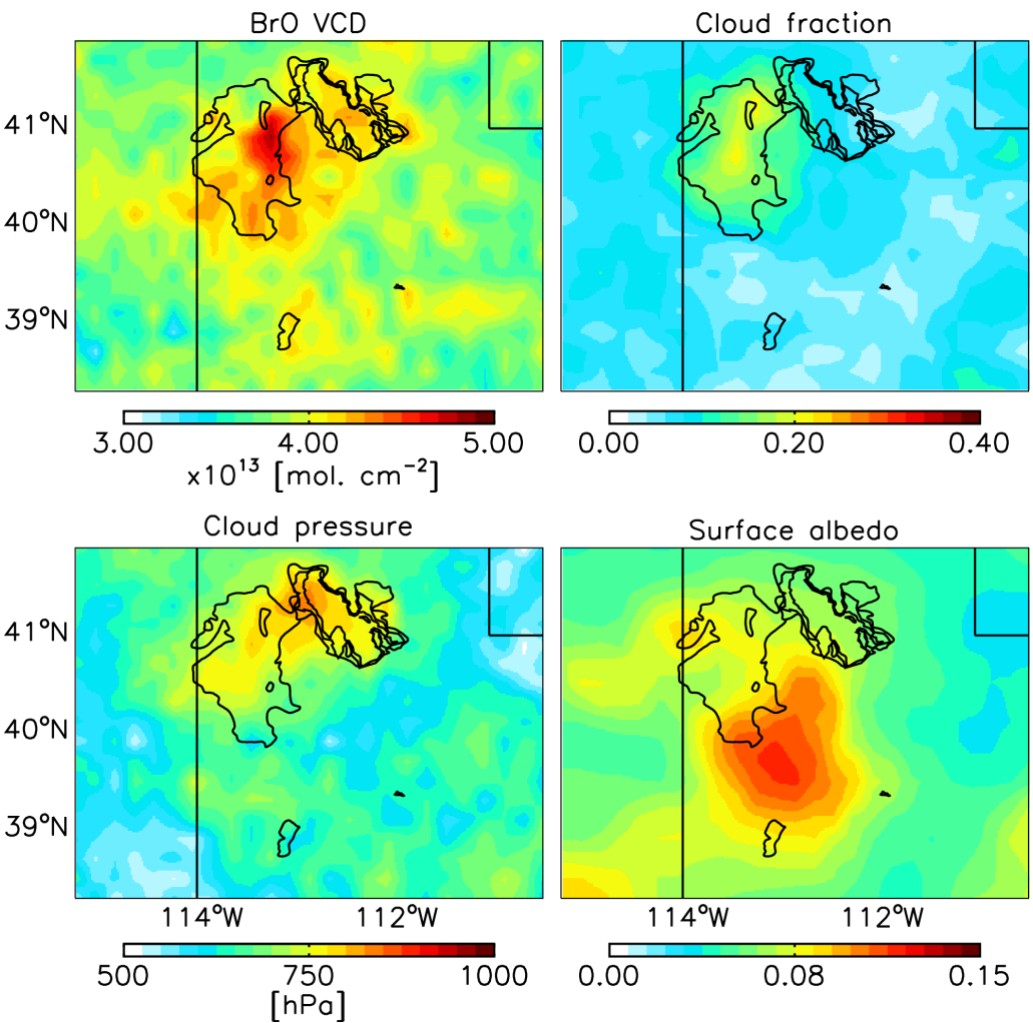

**Figure 11. Mean June 2006 BrO VCD over the Great Salt Lake area. Averages have been calculated on a 0.2° x 0.2° grid including only pixels with cloud fractions smaller than 0.4. The straight lines are borders of the state of Utah, and the curving lines represent the Great Salt Lake (Eastern oval area) and Salt flats (Western oval area).**

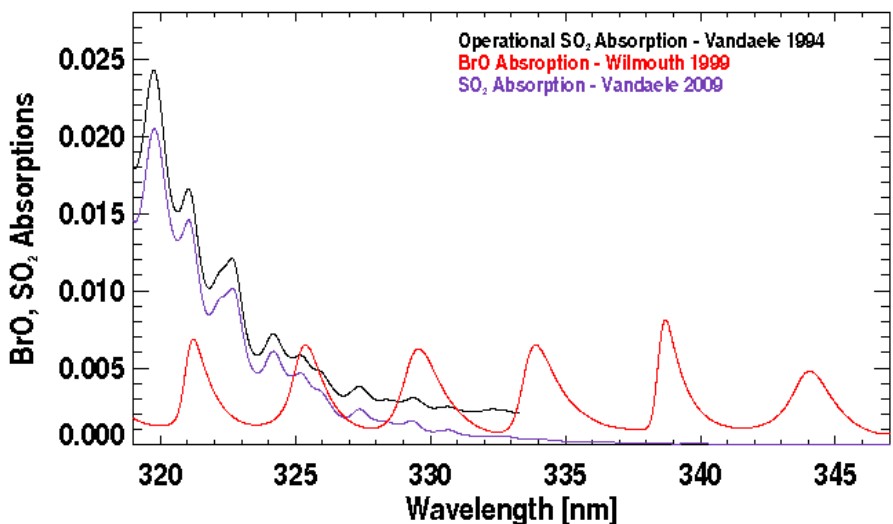

**Figure 12. Comparison of BrO absorption (red) and SO₂ absorptions under volcanic scenarios based on cross sections used in the operational algorithm (Vandaele et al., 1994) as shown in black and the recent laboratory cross sections (Vandaele et al., 2009) as shown in purple. For BrO, a SCD of $1.0 \times 10^{14}$ molecules cm$^{-2}$ is assumed; for SO₂, a SCD of 15 Dobson Units (i.e., $4.03 \times 10^{17}$ molecules cm$^{-2}$) is assumed. Cross sections have been convolved with OMI slit function (which is assumed to be a Gaussian with 0.42nm full width at half maximum).**

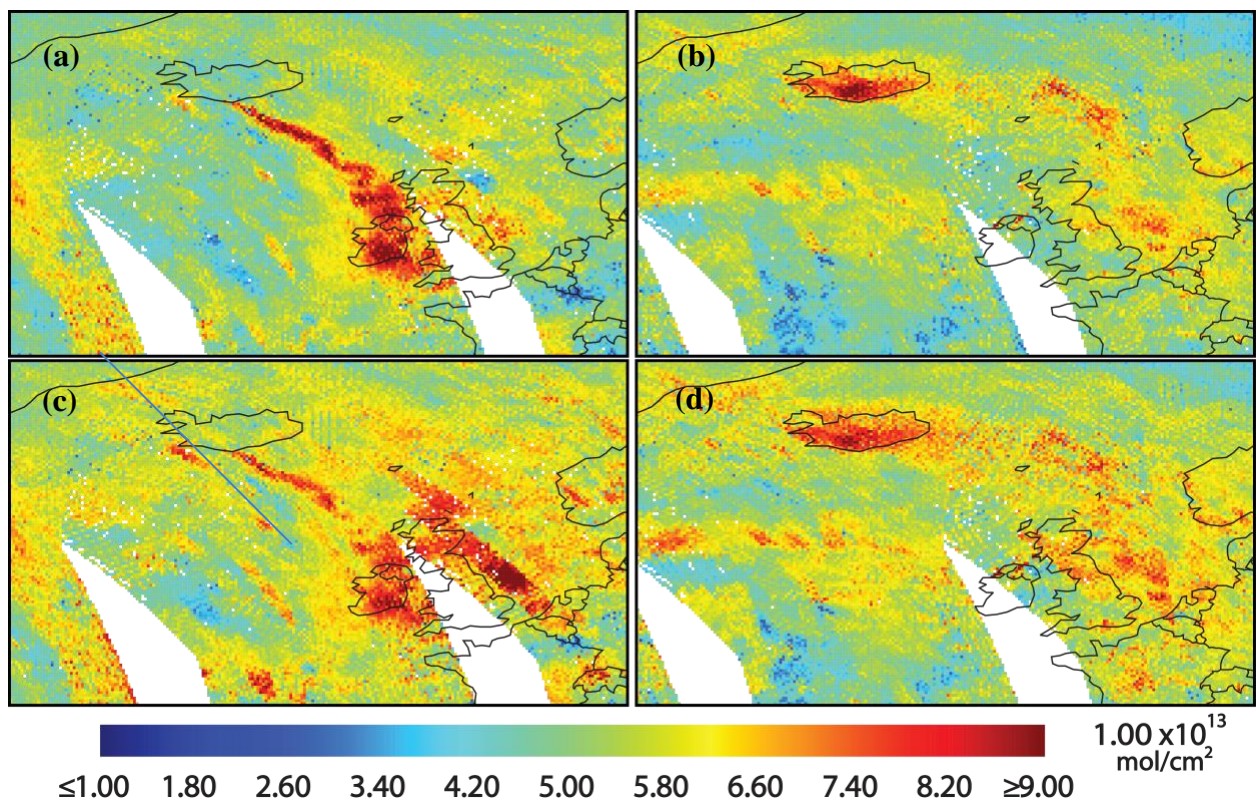

**Figure 13. Daily average BrO VCDs from Eyjafjallajökull on May 5 (a) and 17 (b), 2010 produced using the operational SO₂ cross sections and for the same days (c) and (d) using the Vandaele et al. (2009) SO₂ cross sections.**

