# Peer review of "OMI total bromine monoxide (OMBRO) data product: Algorithm, retrieval and measurement comparisons"

_Atmospheric Measurement Techniques, 2018_

## Referee Comment (RC1) · Anonymous Referee #2 · 14 Feb 2018

**General Comments**

The manuscript gives an overview of the retrieval of BrO VCDs from OMI observations in the OMBRO data product. They then present a comparison of the retrieved VCDs to GOME-2 and ground-based observations at Harestua, Norway, showing general agreement with other BrO observations. Case studies of salt lake observations and volcanic eruptions are also presented, and uncertainties arising from the choice of $SO_2$ cross section are discussed. The topic is appropriate for AMT and the broader community would likely benefit from this publication. However, the presentation of the figures is quite sloppy and some aspects of the main text should be improved prior to

publication. Specific comments are provided below to assist in this process.

**Specific Comments**

Introduction Referencing

I find the choice of references throughout the introduction a bit odd and in some cases not appropriate.

- Page 2, line 6: The knowledge of BrO in the polar troposphere predates both those references by a pretty fair margin. I'd suggest citing some of the earlier observations (e.g. Hausmann and Platt, 1994) or review papers on the topic (e.g. Simpson et al., 2007; Abbatt et al., 2012).

- Page 2, line 7: Hebestreit et al. (1999) should really be cited here.

- Page 2, Line 15: Again, this is a widely studied phenomenon that there are more appropriate references for. See suggested citations in my first comment.

- Page 2, Line 24: If this list is intended to be comprehensive, one should include observations at Alert (e.g. Zhao et al., 2015), Summit, Greenland (Stutz et al., 2011), and throughout the Arctic Ocean (e.g. Burd et al., 2017).

- Page 2, line 26: While many papers have been published on BrO observations at Barrow, Simpson et al. (2005), detailing studies of snowpack chemical composition, is not one of them. Please find a more appropriate reference for this location.
Page 5, Line 8

Remove XtrackQualityFlags and other references to specific data field names throughout the manuscript. In a manuscript it makes more sense to say information is there without referring to a specific field in the data product.

Page 7, Line 16

Specify that the cross sections used can also be found in Table 1

Section 3.6 and 4.4

In my view, the discussion in section 3.6 fits better integrated into section 4.4 since it discusses an application of the data product, not the algorithm itself. Since measurements of halogens in volcanic plumes is a potential use of these data, I think a specific recommendation here would be helpful rather than just advising caution. Would it be an appropriate use of these data to examine BrO production in volcanic plumes?

Page 11, line 21

Since you are comparing 2 sets of satellite observations, orthogonal distance regression would be more appropriate than linear regression here. Linear regression assumes the uncertainty in the GOME2 VCD is much less than that of the OMI VCD, which isn't a valid assumption in this context.
Page 12, line 17

Some context for this correlation would be helpful. How does this correlation compare with other ground-based vs satellite comparisons (e.g. Sihler et al., 2012)?

Page 13, line 28

Provide a reminder of what background values are here.

Suggested Figure Corrections

I'm aware some of these suggestions may seem pedantic, but I found the figure presentation really distracting. The suggested modifications would go a long way toward improving the quality of the manuscript.

- **Figure 4:** Fix y axis label BRO→BrO. Add reference to Operational $SO_2$ and BrO cross section, remove 1st SO2 from Vandaele cross section label. For the sake of consistency, the convoluted Vandaele cross section be shown here rather than the raw laboratory cross section.

- **Figure 5:** Plot against the actual date and explain the large gap in OMI data in the middle of the plot.

- **Figure 6, 8:** These plots are really hard to read. Please consider an alternate font.

- **Figure 7:** Since you are only comparing the total BrO VCD in this work, showing just the time series of the total VCD from Harestua would be more useful than showing three different timeseries from Harestua. You don't really discuss the

other two time series in any meaningful way in the text. Axis labels should also be added.

- **Figure 10:** This should be shown in tandem with a zoomed out map so the reader can orient themselves on the globe and also to show the magnitude of the enhancement relative to the background. The color scale as it currently stands spans a much larger range than that of the data, making it unusable. The map underneath the data is also barely legible.

**Technical Corrections**

Page 1, Line 27

Change "US Great Salt Lake" to the U.S. Great Salt Lake to be consistent with the rest of the manuscript.

Page 3, Line 17

30 pixels?

**References**

Abbatt, J. P. D., Thomas, J. L., Abrahamsson, K., Boxe, C., Granfors, A., Jones, A. E., King, M. D., Saiz-Lopez, A., Shepson, P. B., Sodeau, J., Toohey, D. W., Toubin, C., von Glasow, R., Wren, S. N., and Yang, X.: Halogen activation via interactions with environmental ice and snow in the polar lower troposphere and other regions, Atmospheric Chemistry and Physics, 12, 6237–6271, doi:10.5194/acp-12-6237-2012, http://www.atmos-chem-phys.net/12/6237/2012/, 2012.

Burd, J. A., Peterson, P. K., Nghiem, S. V., Perovich, D. K., and Simpson, W. R.: Snowmelt onset hinders bromine monoxide heterogeneous recycling in the Arctic, Journal of Geophysical Research: Atmospheres, 122, 8297–8309, doi:10.1002/2017JD026906, http://doi.wiley.com/10.1002/2017JD026906, 2017.

Hausmann, M. and Platt, U.: Spectroscopic measurement of bromine oxide and ozone in the high Arctic during Polar Sunrise Experiment 1992, Journal of Geophysical Research, 99, 25 399, doi:10.1029/94JD01314, http://doi.wiley.com/10.1029/94JD01314, 1994.

Hebestreit, K., Stutz, J., Rosen, D., Matveiv, V., Peleg, M., Luria, M., and Platt, U.: DOAS measurements of tropospheric bromine oxide in mid-latitudes, Science, 283, 55–57, doi:10.1126/science.283.5398.55, 1999.

Sihler, H., Platt, U., Beirle, S., Marbach, T., Kühl, S., Dörner, S., Verschaeve, J., Frieß, U., Pöhler, D., Vogel, L., Sander, R., and Wagner, T.: Tropospheric BrO column densities in the Arctic derived from satellite: retrieval and comparison to ground-based measurements, Atmospheric Measurement Techniques, 5, 2779–2807, doi:10.5194/amt-5-2779-2012, http://www.atmos-meas-tech.net/5/2779/2012/, 2012.

Simpson, W. R., Alvarez-Aviles, L., Douglas, T. A., Sturm, M., and Domine, F.: Halogens in the coastal snow pack near Barrow, Alaska: Evidence for active bromine air-snow chemistry during springtime, Geophysical Research Letters, 32, n/a–n/a, doi:10.1029/2004GL021748, http://doi.wiley.com/10.1029/2004GL021748, 2005.

Simpson, W. R., von Glasow, R., Riedel, K., Anderson, P., Ariya, P., Bottenheim, J., Burrows, J., Carpenter, L. J., Frieß, U., Goodsite, M. E., Heard, D., Hutterli, M., Jacobi, H.-W., Kaleschke, L., Neff, B., Plane, J., Platt, U., Richter, A., Roscoe, H., Sander, R., Shepson, P., Sodeau, J., Steffen, A., Wagner, T., Wolff, E., von Glasow, R., Frieß, U., Jacobi, H.-W., Riedel, K., Anderson, P., Ariya, P., Bottenheim, J., Burrows, J., Carpenter, L. J., Frieß, U., Goodsite, M. E., Heard, D., Hutterli, M., Jacobi, H.-W., Kaleschke, L., Neff, B., Plane, J., Platt, U., Richter, A., Roscoe, H., Sander, R., Shepson, P., Sodeau, J., Steffen, A., Wagner, T., and Wolff, E.: Halogens and their role in polar boundary-layer ozone depletion, Atmospheric Chemistry and Physics, 7, 4375–4418, doi:10.5194/acp-7-4375-2007, http://www.atmos-chem-phys.net/7/4375/2007/, 2007.

Stutz, J., Thomas, J. L., Hurlock, S. C., Schneider, M., von Glasow, R., Piot, M., Gorham, K., Burkhart, J. F., Ziemba, L., Dibb, J. E., and Lefer, B. L.: Longpath DOAS observations of surface BrO at Summit, Greenland, Atmospheric Chemistry and Physics, 11, 9899–9910, doi:10.5194/acp-11-9899-2011, http://www.atmos-chem-phys.net/11/9899/2011/

acp-11-9899-2011.html, 2011.

Zhao, X., Strong, K., Adams, C., Schofield, R., Yang, X., Richter, A., Friess, U., Blechschmidt, A. M., and Koo, J. H.: A Case Study of a Transported Bromine Explosion Event in the Canadian High Arctic>, Journal of Geophysical Research: Atmospheres, 121, n/a–n/a, doi:10.1002/2015JD023711, http://onlinelibrary.wiley.com/doi/10.1002/2015JD023711/abstracthttp://doi.wiley.com/10.1002/2015JD023711, 2015.

---

## Referee Comment (RC2) · Anonymous Referee #3 · 3 Apr 2018

**General Comment**

In the paper "OMI total bromine monoxide (OMBRO) data product: Algorithm, retrieval and measurement comparisons" Raid M. Suleiman and co-authors present the operational retrieval algorithm for bromine monoxide (BrO) columns from measurements by the Ozone monitoring instrument (OMI). Since BrO is a trace-gas with significant impact on atmospheric chemistry, the paper fits to the scope of AMT. In my opinion, however, the scientific quality would need more than major revisions because the presentation of the retrieval method in its current form is far from being scientifically publishable. Therefore, I suggest to reject the current manuscript but I would like to

encourage resubmission after the following issues have been addressed.

The decision to suggest the rejection of a manuscript is never easy. In this particular case, however, especial care must be given to scientific quality because operational products are potentially applied by fellow scientists, which may not be trained enough to assess the quality and reliability of the product by themselves. The manuscript, however, rather obfuscates potential quality issues instead of presenting a transparent analysis of the algorithm performance.

**Specific Comments**

1) The most critical aspect of the presented algorithm for the retrieval of BrO from OMI measurements lays in the choice of the wavelength range. The presented algorithm applies a fitting window between 319 to 347.5 nm. Not being an expert for the retrieval of BrO myself, I found the arguments of Vogel et al., 2013 concerning the fit interval of OMBRO particularly alarming. Vogel et al. state in the caption of their Fig. 11 that "Wavelength evaluation ranges with a lower limit <325 are dominated by O3 and SO2 features". In my opinion, this is really alarming since BrO, O3, and SO2 chemistry are highly correlated. The manuscript itself even contains proof that the applied wavelength range might be an issue:

a) Figure 2 shows that the applied AMFs ("OMI current") are structured by O3 absorption indicating that interferences with O3 are close to inevitable.

b) Section 3.6 and Figure 11 reveal interferences with SO2.

c) If scaled properly, Figure 2 would reveal many absorbers to have much larger structures than BrO.

d) Figure 4 is clipped below 330nm. Why? Please show the whole story.

In the current manuscript, however, the choice of the new fitting window is justified in (p.7, l. 2) by the simple statement: "to reduce fitting uncertainty by including more BrO spectral structures". This does not convince me and I really would like to urge the authors to present significant arguments to justify the applied wavelength range, which is far-off compared to the wavelength ranges used by other groups (cf. Table 1 in Vogel et al., 2013). The least the authors could have done would be to include a plot showing results using the "Traditional" and "OMI current" wavelength ranges.

I would like to propose some questions that may lead the authors to find profound arguments: How does the residual change when changing the fitting window? What about systematic structures in the residual? How large are biases by other absorbers depending on the fit range? These questions may be tested using real and synthetic measurements as well using the methods by Chan Miller et al., which was co-authored by many coauthors of this paper. Hence, I wonder why methods for reliably comparing different wavelength ranges were not applied here even though they are existing at SAO. Studies building on this data skate on thin ice if the above issues are not addressed appropriately.

Furthermore (p. 1, l. 18), the authors detail that also "the average fitting residual spectrum" is included in the fit. This approach may obfuscate potential systematic interferences and is, in my opinion, only appropriate if its influence is thoroughly studied. Please provide more information. Please investigate the cross-correlations with other absorbers and include it in a revised Figure 1.

2) It is not clear to me to what extend the presented paper is dedicated to validation of the retrieval results. The title suggests "measurement comparison" and the abstract details that the paper "shows some validation", which is confusing. Please be more specific on the purpose and the results of the validation exercises.

Example: The measurements at Harestua are not ideal for evaluating BrO close to the surface due to a lack of tropospheric BrO events. Therefore, I suggest to state that

the sensitivity of OMBRO towards near-surface BrO may not be evaluated using those measurements. If the authors aim at near-surface BrO with their product, which they do because they claim to have detected BrO over the Great Salt Lake, I suggest to use a data set featuring a significant measurement sensitivity for BrO columns at the ground, for example Frieß et al., 2012.

3) I would like to suggest to review the selection of references in the introduction. There are many citations of papers (co)authored by the coauthors of this paper while papers from other groups seem to be often ignored. E.g. for the sources of tropospheric BrO mostly satellite papers are cited even though there are many observations by groups using ground-based methods. This way of introducing the different findings may be misleading for the readers. Even more, informed fellow scientist readers may be offended if their contribution is not appropriately acknowledged. I suggest to be a bit more generous here. Simpson et al., 2015 may provide a start for a comprehensive list of publications.

Specifically, I miss the following references in an up-to-date BrO satellite paper:

- Please add (Hörmann et al., 2013), which is one of the most comprehensive surveys of volcanic BrO sources using satellites.

- Please add (Liao et al., 2011) and (Frieß et al., 2012) to the references for Barrow, Alaska since both papers present significant BrO observations of near-surface BrO.

- Further BrO satellite papers well worth citing: Begoin et al., 2010, Toyota et al., 2011, Sihler et al., 2012, and Blechschmidt et al., 2016

4) The investigation of the BrO over the Great Salt lake is insufficient and I am missing a rationale for including this issue in the paper at all. However, the results may be due to systematic effects caused by a variety of geophysical parameters (see investigation by Hörmann et al., 2016). Without an appropriate discussion of these influences I would not accept the authors claim that the signal is really due to emissions from the Great

[Figure]

Salt Lake.

5) In my opinion, the treatment of the interferences with SO2 is not appropriate. If the issue is known, why not solve it right-away and then publish an improved version? What is the purpose of an OMBRO product featuring this imperfection? I suggest to solve this issue together with choosing an appropriate fitting window before resubmission.

Specifically:

a) Fig. 11 is really hard to interpret. I suggest to show a comparison plot based on single OMI measurements instead of gridded maps.

b) In the introduction (p.2, l.31): Please rephrase "a known issue" for something more specific. In my opinion, the interference with SO2 is not just an issue but a significant flaw.

c) Fig. 4: The x-axis must contain the entire fitting window at least. I find the figure in its current form rather disturbing.

6) The following plots need to be improved:

a) Fig. 1: I strongly suggest to refrain from scaling arbitrarily to allow an open discussion of the results. For example, the amplitude of O3 and SO2 cross sections and the Ring spectra seem to be strongly manipulated in order to downplay their potential impact. I suggest to apply an y-axis in optical density space and scale the cross-sections according to a typical fit.

b) Fig. 6: Please use orthogonal regression for the comparison. Linear regression is not appropriate for independent data sets.

c) Fig. 7: The frequency of the time series is too high to allow a one-to-one comparison. I recommend to also show a zoomed plot of two months or so.

d) Fig. 8: see 6c)

e) Fig. 10: This plot does not allow an independent judgment whether this is a significant signal or not. Suggested improvements:

- Increase area significantly

- Use full colorscale

- Thicker coast lines

- Align with other geospatial properties: cloud statistics, albedo, precipitation etc.

**Further Comments**

(p. 5, l. 16) "Unlike the often-used DOAS fitting method (Platt, 1994), radiances are not ratioed to irradiances, logarithms are not taken, and no high-pass filtering is applied." I wonder whether this is an advantage or disadvantage of the described method. What is the intention behind this statement? My suggestion would be not to confuse the reader and just remove it from the manuscript.

(p. 11, l.17) Why are OMI and GOME-2 data treated differently with respect to spatial averaging? Without discussion, the reader may assume that OMI data are more noisy and needed some smoothing. Please be more specific.

(p. 13, l.21 -> l. 25) -> move to introduction

(p. 14, l. 2 -> l. 99 -> move to introduction

(p.2, l. 29): "briefly analyze" characterizes an approach not suitable for a scientific article. An analysis is either profound or not scientific. In my opinion, an AMT paper should only contain profound content.

(p.3, l. 22): Please add reference documenting the OMI row anomaly: http://projects.knmi.nl/omi/research/product/rowanomaly-background.php

**1 References**

M. Begoin, A. Richter, M. Weber, L. Kaleschke, X. Tian-Kunze, A. Stohl, N. Theys, and J. P. Burrows: Satellite observations of long range transport of a large BrO plume in the Arctic Atmos. Chem. Phys., 10, 6515-6526, doi:10.5194/acp-10-6515-2010, 2010.

K. Toyota, J. C. McConnell, A. Lupu, L. Neary, C. A. McLinden, A. Richter, R. Kwok, K. Semeniuk, J. W. Kaminski, S.-L. Gong, J. Jarosz, M. P. Chipperfield, and C. E. Sioris Atmos. Chem. Phys., 11, 3949-3979, doi:10.5194/acp-11-3949-2011, 2011.

U. Frieß, H. Sihler, R. Sander, D. Pohler, S. Yilmaz, and U. Platt: The vertical distribution of BrO and aerosols in the Arctic: measurements by active and passive differential optical absorption sprectroscopy J. Geophys. Res., 116, D00R04, doi:10.1029/2011JD015938, 2011.

J. Liao, H. Sihler, L. G. Huey, J. A. Neuman, D. J. Tanner, U. Friess, U. Platt, F. M. Flocke, J. J. Orlando, P. B. Shepson, H. J. Beine, A. J. Weinheimer, S. J. Sjostedt, J. B. Nowak, D. J. Knapp, R. M. Staebler, W. Zheng, R. Sander, S. R. Hall, and K. Ullmann: A comparison of Arctic BrO measurements by chemical ionization mass spectrometry and long path-differential optical absorption spectroscopy J. Geophys. Res.-Atmos., 116, D00r02, doi:10.1029/2010jd014788, 2011.

H. Sihler, U. Platt, S. Beirle, T. Marbach, S. Kühl, S. Dörner, J. Verschaeve, U. Frieß, D. Pöhler, L. Vogel, R. Sander, and T. Wagner: Tropospheric BrO column densities in the Arctic derived from satellite: retrieval and comparison to ground-based measurements Atmos. Meas. Tech., 5, 2779-2807, doi:10.5194/amt-5-2779-2012, 2012.

L. Vogel, H. Sihler, J. Lampel, T. Wagner, and U. Platt: Retrieval interval mapping: a tool to visualize the impact of the spectral retrieval range on differential optical absorption spectroscopy evaluations Atmos. Meas. Tech., 6, 275-299, doi:10.5194/amt-6-275-2013, 2013.

C. Hörmann, H. Sihler, N. Bobrowski, S. Beirle, M. Penning de Vries, U. Platt, and T. Wagner: Systematic investigation of bromine monoxide in volcanic plumes from space by using the GOME-2 instrument Atmos. Chem. Phys., 13, 4749-4781, doi:10.5194/acp-13-4749-2013, 2013.

C. Chan Miller, G. Gonzalez Abad, H. Wang, X. Liu, T. Kurosu, D. J. Jacob, and K. Chance: Glyoxal retrieval from the Ozone Monitoring Instrument Atmos. Meas. Tech., 7, 3891-3907, doi:10.5194/amt-7-3891-2014, 2014.

W. Simpson, S. Brown, A. Saiz Lopez, J. Thornton, R. von Glasow: Tropospheric halogen chemistry: sources, cycling, and impacts Chemical Reviews, 115, 10, 4035-4062, doi:10.1021/cr5006638, 2015.

A.-M. Blechschmidt, A. Richter, J. P. Burrows, L. Kaleschke, K. Strong, N. Theys, M. Weber, X. Zhao, and A. Zien: An exemplary case of a bromine explosion event linked to cyclone development in the Arctic Atmos. Chem. Phys., 16, 1773-1788, doi:10.5194/acp-16-1773-2016, 2016.

C. Hörmann, H. Sihler, S. Beirle, M. Penning de Vries, U. Platt, and T. Wagner Atmos. Chem. Phys., 16, 13015-13034, doi:10.5194/acp-16-13015-2016, 2016.
* * *

---

## Author Comment (AC1) · 14 May 2018

We thank referee's helpful and constructive comments and review. Our responses are in **bold** starting with **"Response:"**

**Figures have been moved around:**
**Old manuscript Updated manuscript**
**Figure 1 still Figure 1**
**Figure 2 still Figure 2**
**Figure 3 still Figure 3**
**Figure 4 moved now Figure 10**

**Figure 5 moved now Figure 4**
**Figure 6 moved now Figure 5**
**Figure 7 moved now Figure 6**
**Figure 8 moved now Figure 7**
**Figure 9 moved now Figure 8**
**Figure 10 moved now Figure 9**
**Figure 11 still Figure 11**

**Please note: all mention of figure 's below corresponds to the updated numbering in the updated manuscript.**
**Please note that many figures have been changed based on another referee's suggestions.**
**- Figure 1 plotted as a function of optical thickness**
**- Figure 4 was redone without averaging OMI BrO data and a new orthogonal regression was performed (Figure 5)**
**- Section 3.6 was moved to 4.4**
**- Figure 9. We increased the covered area, used a full-color scale, and added the lake line.**
**- Added another panel to Figure 9 to show BrO over the Dead Sea Valley for September 2007.**
**- Figure 10 was extended to cover the full wavelength window**

Anonymous Referee 2 General Comments The manuscript gives an overview of the retrieval of BrO VCDs from OMI observations in the OMBRO data product. They then present a comparison of the retrieved VCDs to GOME-2 and ground-based observations at Harestua, Norway, showing general agreement with other BrO observations. Case studies of salt lake observations and volcanic eruptions are also presented, and uncertainties arising from the choice of $SO_2$ cross section are discussed. The

topic is appropriate for AMT and the broader community would likely benefit from this publication. However, the presentation of the figures is quite sloppy and some aspects of the main text should be improved prior to publication. Specific comments are provided below to assist in this process.

Specific Comments
Introduction Referencing
I find the choice of references throughout the introduction a bit odd and in some cases not appropriate.

Page 2, line 6: The knowledge of BrO in the polar troposphere predates both those references by a pretty fair margin. I'd suggest citing some of the earlier observations (e.g. Hausmann and Platt, 1994) or review papers on the topic (e.g. Simpson et al., 2007; Abbatt et al., 2012).

**Response:**
**Added the suggested references.**

Page 2, line 7: Hebestreit et al. (1999) should really be cited here. Response: Added "Hebestreit, et al., 1999"

Page 2, Line 15: Again, this is a widely studied phenomenon that there are more appropriate references for. See suggested citations in my first comment.

**Response:**
**More references were added.**

(Hausmann and Platt, 1994; von Glasow et al., 2004; Salawitch et al., 2005; Simpson et al., 2007; Salawitch et al., 2010; Abbatt, et al., 2012).

Page 2, Line 24: If this list is intended to be comprehensive, one should include observations at Alert (e.g. Zhao et al., 2015), Summit, Greenland (Stutz et al., 2011), and throughout the Arctic Ocean (e.g. Burd et al., 2017).

**Response:**
**Added the suggested references.**
**The manuscript changed to: and Barrow, Alaska (Liao et al., 2012a,b; Frieß et al., 2011; Sihler et al., 2012; Peterson et al., 2016), Eureka, Canada (Zhao et al., 2015), Summit, Greenland (Stutz et al., 2011) and the Arctic Ocean (Burd et al., 2017).**

Page 2, line 26: While many papers have been published on BrO observations at Barrow, Simpson et al. (2005), detailing studies of snowpack chemical composition, is not one of them. Please find a more appropriate reference for this location.

**Response:**
**Removed Simpson et al. (2005) and added two appropriate references. The new text "and Barrow, Alaska (Frieß et al., 2011; Liao et al., 2012a,b; Sihler et al., 2012; Peterson et al., 2016),"**

Page 5, Line 8
Remove XtrackQualityFlags and other references to specific data field names throughout the manuscript. In a manuscript it makes more sense to say information is there without referring to a specific field in the data product.

[Figure]

**Response:**
**Removed XtrackQualityFlags**

Page 7, Line 16
Specify that the cross sections used can also be found in Table 1

**Response:**
**Modified the sentence to be "The operational parameters and the cross sections used are provided in Table 1."**

Section 3.6 and 4.4
In my view, the discussion in section 3.6 fits better integrated into section 4.4 since it discusses an application of the data product, not the algorithm itself. Since measurements of halogens in volcanic plumes is a potential use of these data, I think a specific recommendation here would be helpful rather than just advising caution. Would it be an appropriate use of these data to examine BrO production in volcanic plumes?

**Response:**
**Section 3.6 has been moved and incorporated into section 4.4.**

Page 11, line 21
Since you are comparing 2 sets of satellite observations, orthogonal distance regression would be more appropriate than linear regression here. Linear regression assumes the uncertainty in the GOME2 VCD is much less than that of the OMI VCD, which isn't a valid assumption in this context.

**Response:**
**We corrected Figure 5 using orthogonal distance regression.**

Page 12, line 17
Some context for this correlation would be helpful. How does this correlation compare
with other ground-based vs satellite comparisons (e.g. Sihler et al., 2012)?

**Response:**
**Added to the manuscript:**
**Sihler et al. (2012) compared GOME-2 BrO to ground-based observations at**
**Barrow finding the correlation to be weaker (r = 0.3), likely due to both elevated**
**and shallow surface layers of BrO. However, their correlation between GOME-2**
**BrO and ground-based measurements at Amundsen, U.S. (r = 0.4) is closer to**
**our correlation here.**

Page 13, line 28
Provide a reminder of what background values are here.

**Response:**
**In the manuscript we replaced:**
**BrO enhancement of 5-10$\times$1012 molecules cm-2 over background values is**
**clearly shown right over this salt lake.**
**with:**
**Over the Great Salt Lake, BrO enhancement occurs predominantly over the lake**
**bed with enhancements of 5-10$\times$1012 molecules cm-2 over background values**
**(4-4.7$\times$1013 molecules cm-2).**
**Please note that we have included discussion and a plot of BrO enhancement**

**over the Dead Sea Valley from 09/2007.**

Suggested Figure Corrections
I'm aware some of these suggestions may seem pedantic, but I found the figure presentation really distracting. The suggested modifications would go a long way toward improving the quality of the manuscript.

• Figure 4: Fix y axis label BRO→BrO. Add reference to Operational SO2 and BrO cross section, remove 1st SO2 from Vandaele cross section label. For the sake of consistency, the convoluted Vandaele cross section be shown here rather than the raw laboratory cross section.

Added to the end of Figure 10 caption:
Cross sections have been convolved with OMI slit function (which is assumed to be a Gaussian with 0.42nm full width at half maximum).**Response:**
**Old Figure 4 (now Figure 10) was updated to include the entire fitting window, the references for the cross sections. Additionally, the updated Figure 10 now shows the cross sections after they have been convolved with OMI slit function (which is assumed to be a Gaussian with 0.42nm).**

**Added to the end of Figure 10 caption:**
**Cross sections have been convolved with OMI slit function (which is assumed to be a Gaussian with 0.42nm full width at half maximum).**
**Figure 5: Plot against the actual date and explain the large gap in OMI data in the middle of the plot.**

**Response:**
**We updated old Figure 5 (now Figure 4) so the x-axis is now the actual date. The gap in OMI data is due to the filtering of retrievals with bad quality flags. We also did not average OMI data anymore by using individual OMI pixels and we relaxed the quality flag selection which eliminated the gap.**

**Figure 6, 8: These plots are really hard to read. Please consider an alternate font.**
**Response:**
**Figure 5 and 7 have been updated to use orthogonal regression and used a more suitable font.**
**Changed caption of Figure 5:**
**Correlation and orthogonal regression of OMI and GOME-2 BrO for the data in Fig. 4 when both data are available. The legends show the mean biases and standard deviations of the differences, correlation, and the orthogonal regression.**
**Changed the caption of Figure 7:**
**Correlation and orthogonal regression of OMI and Harestua BrO for the data in Fig. 6. The legends show the mean biases and standard deviations of the differences, correlation, and the orthogonal regression.**
**Figure 7: Since you are only comparing the total BrO VCD in this work, showing just the time series of the total VCD from Harestua would be more useful than showing three different timeseries from Harestua. You don't really discuss the other two time series in any meaningful way in the text. Axis labels should also be added.**

**Response:**
**Figure 6 have been updated to include total BrO only at Harestua. We also added the axis labels.**

Figure 10: This should be shown in tandem with a zoomed out map so the reader can orient themselves on the globe and also to show the magnitude of the enhancement relative to the background. The color scale as it currently stands spans a much larger range than that of the data, making it unusable. The map underneath the data is also barely legible.

Response:
Updated Figure 9 as suggested and added another panel to show BrO over the Dead Sea Valley.

Technical Corrections
Page 1, Line 27 Change "US Great Salt Lake" to the U.S. Great Salt Lake to be consistent with the rest of the manuscript.

Response:
Changed US to U.S.

Page 3, Line 17 30 pixels?

Response:
Added the word "pixels" after 30

---

## Author Comment (AC2) · 14 May 2018

We thank referee's helpful and constructive comments and review. Our responses are in **bold** starting with **"Response:"**
**Figures have been moved around:**
**Old manuscript Updated manuscript**
**Figure 1 still Figure 1**
**Figure 2 still Figure 2**
**Figure 3 still Figure 3**
**Figure 4 moved to become Figure 10**

[Figure]

**Figure 5 moved to become Figure 4**
**Figure 6 moved to become Figure 5**
**Figure 7 moved to become Figure 6**
**Figure 8 moved to become Figure 7**
**Figure 9 moved to become Figure 8**
**Figure 10 moved to become Figure 9**
**Figure 11 still Figure 11**

General Comment

In the paper "OMI total bromine monoxide (OMBRO) data product: Algorithm, retrieval and measurement comparisons" Raid M. Suleiman and co-authors present the operational retrieval algorithm for bromine monoxide (BrO) columns from measurements by the Ozone monitoring instrument (OMI). Since BrO is a trace-gas with significant impact on atmospheric chemistry, the paper fits to the scope of AMT. In my opinion, however, the scientific quality would need more than major revisions because the presentation of the retrieval method in its current form is far from being scientifically publishable. Therefore, I suggest to reject the current manuscript but I would like to encourage resubmission after the following issues have been addressed. The decision to suggest the rejection of a manuscript is never easy. In this particular case, however, especial care must be given to scientific quality because operational products are potentially applied by fellow scientists, which may not be trained enough to assess the quality and reliability of the product by themselves. The manuscript, however, rather obfuscates potential quality issues instead of presenting a transparent analysis of the algorithm performance.

Specific Comments

1) The most critical aspect of the presented algorithm for the retrieval of BrO from OMI measurements lays in the choice of the wavelength range. The presented algorithm applies a fitting window between 319 to 347.5 nm. Not being an expert for the retrieval of BrO myself, I found the arguments of Vogel et al., 2013 concerning the fit interval of OMBRO particularly alarming. Vogel et al. state in the caption of their Fig. 11 that

"Wavelength evaluation ranges with a lower limit <325 are dominated by O3 and SO2 features". In my opinion, this is really alarming since BrO, O3, and SO2 chemistry are highly correlated. The manuscript itself even contains proof that the applied wavelength range might be an issue:

a) Figure 2 shows that the applied AMFs ("OMI current") are structured by O3 absorption indicating that interferences with O3 are close to inevitable.

b) Section 3.6 and Figure 11 reveal interferences with SO2.

c) If scaled properly, Figure 2 would reveal many absorbers to have much larger structures than BrO.

d) Figure 4 is clipped below 330nm. Why? Please show the whole story

In the current manuscript, however, the choice of the new fitting window is justified in (p.7, l. 2) by the simple statement: "to reduce fitting uncertainty by including more BrO spectral structures". This does not convince me and I really would like to urge the authors to present significant arguments to justify the applied wavelength range, which is far-off compared to the wavelength ranges used by other groups (cf. Table 1 in Vogel et al., 2013). The least the authors could have done would be to include a plot showing results using the "Traditional" and "OMI current" wavelength ranges. I would like to propose some questions that may lead the authors to find profound arguments: How does the residual change when changing the fitting window? What about systematic structures in the residual? How large are biases by other absorbers depending on the fit range? These questions may be tested using real and synthetic measurements as well using the methods by Chan Miller et al., which was co-authored by many coauthors of this paper. Hence, I wonder why methods for reliably comparing different wavelength ranges were not applied here even though they are existing at SAO. Studies building on this data skate on thin ice if the above issues are not addressed appropriately.

Furthermore (p. 1, l. 18), the authors detail that also "the average fitting residual spectrum" is included in the fit. This approach may obfuscate potential systematic interferences and is, in my opinion, only appropriate if its influence is thoroughly

studied. Please provide more information. Please investigate the cross-correlations with other absorbers and include it in a revised Figure 1.

**Response:**

**We first proposed that BrO could be measured globally from satellites (Chance, K.V., J.P. Burrows, and W. Schneider, Retrieval and molecule sensitivity studies for the Global Ozone Monitoring Experiment and the SCanning Imaging Absorption spectroMeter for Atmospheric CHartographY, Proc. SPIE, Remote Sensing of Atmospheric Chemistry, 1491, 151-165, 1991; http://www.cfa.harvard.edu/atmosphere/publications.html). We were then the first to fit BrO from GOME-1 and have fitted BrO from SCIAMACHY, OMI, and OMPS. The update of the BrO fitting window from V2 to V3 in 2011 is not arbitrary, but is based on substantial quantitative analysis by checking the quality of BrO retrievals and the correlation with other trace gases while systematically varying the lower and upper limits of fitting windows, similar to the studies of Chan Miller et al. (2014). It is obvious that O3 absorption and SO2 absorption are much stronger than BrO absorption below 325 nm, but it does not mean BrO cannot be retrieved using part of this wavelength range as we are simultaneously fitting O3 and SO2. For example, operational SO2 measurements from UV are almost always retrieved from a window that is entirely dominated by the O3 absorption for typical SO2 abundance. We state in this current paper that "The correlation of the unmodified BrO cross sections with the rest of the molecules fitted is small (typically less than 0.12), except with H2CO (0.43). However, it is safe to assume that in most polar regions with enhanced BrO there are no high concentrations of formaldehyde."**

**The caption to Fig. 11 in Vogel et al., (2013) states that "wavelength evaluation ranges with a lower limit <325 are dominated by O3 and SO2 features, whereas the other wavelength ranges may be influenced mainly by NO2 and HCHO." This is a comparison of where different molecules might interfere based**

on the absorption not an absolute determination of the best range. Our careful evaluation of correlation stands. We have been very systematic, as in Chan Miller et al., 2014, in the selection of our wavelength range. If one puts a point on Fig. 11 of Vogel et al. (2013) corresponding to our fitting window, one would see that the correlation is not significant for our measurements. In fact, Vogel et al. (2013) stated in their Appendix B conclusion (B3) that "If the retrieval wavelength interval is sufficiently wide and includes strong BrO absorption lines, the results are not strongly affected by changes in retrieval wavelength intervals (lower wavelength limits 320–337.5 nm and upper wavelength limit > 342 nm)."

"a) Figure 2 shows that the applied AMFs ("OMI current") are structured by O3 absorption indicating that interferences with O3 are close to inevitable."

Yes. AMFs should be affected by the dominant O3 absorption. We agree that interferences with O3 is close to inevitable if O3 is not fitted. We simultaneously fit O3 and account for this wavelength-dependent AMF to minimize the O3 interference so the correlations between BrO and O3 (218 K) and O3 (298 K) are typically very small, less than 0.03

"b) Section 3.6 and Figure 11 reveal interferences with SO2."

This is fully explained in Section 4.3 second paragraph (starting on line 15 of the updated manuscript). Correlation with volcanic SO2 will be reduced when improved cross sections are used. Correlation with normal SO2 is negligible given the low levels of SO2 present in the atmosphere. As we point out in the volcano detection section, when high levels of SO2 are present OMI BrO should be utilized with extra care.

"c) If scaled properly, Figure 2 would reveal many absorbers to have much larger structures than BrO."

**Looks like you meant Figure 1. This is why spectra must be fitted very carefully and correlations evaluated. Otherwise, nobody could fit atmospheric BrO or most other trace gases in the UV/visible. We have changed the approach to Figure 1 by showing four different panels with 4 different optical density orders of magnitude.**

"d) Figure 4 is clipped below 330nm. Why? Please show the whole story."

**We are making no attempts to hide information. On the contrary, we use this wavelength range to highlight the region where the new and old SO2 cross sections differ the most. We have modified the figure to show the whole fitting window.**

**In summary, we urge the referee to discount Comment 1 arguments for rejection and accept our paper. We consider that the goal of the paper, to provide a scientific description of SAO OMI BrO retrievals and some particular examples, is achieved. We appreciate the comments that helped us to improve the clarity and transparency of the manuscript. However, we want to clearly reject the idea, as hinted throughout the comments, that we are making an intentional effort to keep the paper obscure.**

2) It is not clear to me to what extend the presented paper is dedicated to validation of the retrieval results. The title suggests "measurement comparison" and the abstract details that the paper "shows some validation", which is confusing. Please be more specific on the purpose and the results of the validation exercises. Example: The measurements at Harestua are not ideal for evaluating BrO close to the surface due to

a lack of tropospheric BrO events. Therefore, I suggest to state that the sensitivity of OMBRO towards near-surface BrO may not be evaluated using those measurements. If the authors aim at near-surface BrO with their product, which they do because they claim to have detected BrO over the Great Salt Lake, I suggest to use a data set featuring a significant measurement sensitivity for BrO columns at the ground, for example Frieb et al., 2012.

**Response:**
**The purpose of the paper is not to do a dedicated validation exercise but to describe the algorithm and show some comparisons, consistent with the title. We show some comparisons of total BrO with GOME-2 and ground-based observations, and show examples of BrO enhancement from volcanic eruptions and salt lakes. We are not trying to evaluate BrO close to the surface or in the troposphere as the retrieval currently assumes a mostly stratospheric BrO profile. But, this does not prevent us from showing enhanced BrO due to sources near the surface as these sources will contribute to the total BrO although total BrO will be underestimated due to the assumed BrO profile. In the abstract, we replaced "shows some validation" with "comparisons."**

3) I would like to suggest to review the selection of references in the introduction. There are many citations of papers (co)authored by the coauthors of this paper while papers from other groups seem to be often ignored. E.g. for the sources of tropospheric BrO mostly satellite papers are cited even though there are many observations by groups using ground-based methods. This way of introducing the different findings may be misleading for the readers. Even more, informed fellow scientist readers may be offended if their contribution is not appropriately acknowledged. I suggest to be a bit more generous here. Simpson et al., 2015 may provide a start for a comprehensive list of publications.
Specifically, I miss the following references in an up-to-date BrO satellite paper:

- Please add (Hörmann et al., 2013), which is one of the most comprehensive surveys of volcanic BrO sources using satellites.

**Response:**
**We added this reference in the revision with the text: "Hörmann et al. (2013) examined GOME-2 observations of BrO slant column densities (SCDs) in the vicinity of volcanic plumes; it showed clear enhancements of BrO in 1/4 of the volcanos, and revealed large spatial differences in BrO/SO2 ratios."**

- Please add (Liao et al., 2011) and (Frieß et al., 2012) to the references for Barrow, Alaska since both papers present significant BrO observations of near-surface BrO.

**Response:**
**The recommended two references were added. Please note that Liao, et al., 2011 was finally published in 2012 and Frieß was published in 2011 not 2012.**
**Text added to manuscript: Frieß et al., 2011; Liao et al., 2012a,b;**

- Further BrO satellite papers well worth citing: Begoin et al., 2010, Toyota et al., 2011, Sihler et al., 2012, and Blechschmidt et al., 2016

**Response:**
**Begoin et al. (2010) was added in the Introduction.**
**Toyota et al. (2011) is a modeling paper and it has no observations or comparison to any satellite or ground-based measurements, so it is not added.**
**Sihler et al. (2012) and Blechschmidt et al. (2016) were added to the Introduction.**

4) The investigation of the BrO over the Great Salt lake is insufficient and I am missing a rationale for including this issue in the paper at all. However, the results may be due to systematic effects caused by a variety of geophysical parameters (see investigation by Hörmann et al., 2016). Without an appropriate discussion of these influences I would not accept the authors claim that the signal is really due to emissions from the Great Salt Lake.

**Response:**
**We have added several sentences at the beginning of section 4 to show the rationale "Comparisons of the OMI OMBRO product with GOME-2 satellite retrievals and remote sensing ground-based measurements over Harestua, Norway as well as monthly mean averages illustrate the quality of the retrieval on a global scale. On a local scale recent scientific studies looking at BrO enhancements in volcanic plumes and over salt lakes are pushing the limits of the current OMBRO setups. In the following sections we provide details of these comparisons (section 4.1) and discuss OMI OMBRO global distribution (section 4.2) and local enhancements over salt lakes and volcanic plumes observations (section 4.3), and their applicability and strategies to correctly use the publicly available OMBRO product."**

**The paragraph has been significantly updated. In addition to BrO enhancement over the Great Salt Lake, we added BrO enhancement also over the Dead Sea Valley. The impacts of geophysical parameters were discussed. This is the updated text:**

**Following recent work by Hörmann et al. (2016) we have checked the capability of OMBRO to observe similar enhancements in other salt lakes. Fig. 9 shows monthly averaged OMI BrO over the Great Salt Lake for 02/2013.**

and the Dead Sea for 07/2009. Over the Great Salt Lake, BrO enhancement occurs predominantly over the lake bed with enhancements of  5-10×1012 molecules cm-2 over background values (4-4.7×1013 molecules cm-2). Over the Dead Sea, the BrO enhancement of 5-8 ×1012 molecules cm-12 occurs to the South-West, where BrO accumulates at a small hill due to the prevailing north-easterly winds. Despite observing these enhancements, the users of OMBRO for these kinds of studies should be aware of two limitations of the current retrieval. First, the actual BrO enhancement is actually underestimated since we are assuming a mostly stratospheric BrO profile for the AMF. Second, the OMI derived albedo climatology (Kleipool et al. 2008) used in OMBRO has a resolution of 0.5 degrees. At this resolution OMBRO retrievals can have biases given the size of OMI pixels and sub-pixel albedo variability not represented in the albedo climatology. We also raise attention to the fact that abnormally high cloud fractions are reported over the salt lakes due to enhanced albedos. All these considerations are important for future studies studying spatiotemporal distribution of BrO over salt lakes.

5) In my opinion, the treatment of the interferences with SO2 is not appropriate. If the issue is known, why not solve it right-away and then publish an improved version? What is the purpose of an OMBRO product featuring this imperfection? I suggest to solve this issue together with choosing an appropriate fitting window before resubmission.

**Response:**
**All operational algorithms of all gases contain known issues. There are usually suggestions as to how future improvements can be made and how to use the data properly. Interference with SO2 is only an issue when SO2 concentrations become significant. Keeping that in mind the main purpose of discussing the impact of the SO2 cross sections in the retrieval is to educate possible users**

**about the limitations of the current set up in specific scenarios (i.e. volcanic plumes and smelters). We discuss the effect of the new cross sections (Vandaele et al., 2009) since we plan to include them in future updates to the operational processing. We also look forward to laboratory measurements over an extended temperature range, and have encouraged them. However, given the limited impact in the retrieval results (from a geographic and occurrence perspective), the complications and limitations arising from operational processing and the computing cost of full mission reprocessing we prefer to apply several updates in each new version. With plans to update other aspects of the retrieval we favor the option of saving the SO2 cross section update for the future while we still think is important to single out the limitations of the current setup so that all known limitations of the OMI BrO are available to the science community in order to make educated decisions on which kind of science investigations are supported by the data product, and which are not. This is completely standard in the development of operational products.**

Specifically:
a) Fig. 11 is really hard to interpret. I suggest to show a comparison plot based on single OMI measurements instead of gridded maps.

**Response:**
**A plot of a single OMI measurement does not add much. Due to significant orbit overlapping at this location. The results need to be averaged to show entire structures clearly. For reference, two separate orbits are shown below (Figures 1-4).**

b) In the introduction (p.2, l.31): Please rephrase "a known issue" for something more specific. In my opinion, the interference with SO2 is not just an issue but a significant flaw.

**Response:**
**As explained above, it is not a significant flaw, nor another reason to reject the paper. We have changed "a known issue" to SO2 interference.**

c) Fig. 4: The x-axis must contain the entire fitting window at least. I find the figure in its current form rather disturbing.

**Response:**
**"Rather disturbing" is not appropriate language for an unbiased scientific review. We use this wavelength range to highlight the region where the new and old SO2 cross sections differ the most. Figure 4 (in the updated manuscript moved to be Figure 10) is updated to include the entire fitting window. Additionally, we plotted cross sections that have been convolved with OMI slit function (which is assumed to be a Gaussian with 0.42nm full width at half maximum).**

6) The following plots need to be improved:

a) Fig. 1: I strongly suggest to refrain from scaling arbitrarily to allow an open discussion of the results. For example, the amplitude of O3 and SO2 cross sections and the Ring spectra seem to be strongly manipulated in order to downplay their potential impact. I suggest to apply an y-axis in optical density space and scale the cross-sections according to a typical fit.

**Response:**
**It was not intentional to downplay any contribution from any cross sections through arbitrary scaling. Our intention was to provide a qualitative image of the species involved in the fitting while keeping the figure simple. We have changed the approach to Figure 1 by showing four different panels with 4 different optical**

**density orders of magnitude.**

b) Fig. 6: Please use orthogonal regression for the comparison. Linear regression is not appropriate for independent data sets.

**Response:**
**We corrected old Figure 6 (now Figure 5) using orthogonal distance regression.**

**Changed caption of old Figure 6:**
**Correlation and orthogonal regression of OMI and GOME-2 BrO for the data in Figure 4 when both are available. The legends show the mean bias and standard deviations of the differences, correlation, and the orthogonal regression.**

c) Fig. 7: The frequency of the time series is too high to allow a one-to-one comparison. I recommend to also show a zoomed plot of two months or so.

**Response:**
**Old Figure 7 (now Figure 6) have been updated to include only the time series from Harestua and OMI total BrO only since we are not discussing in the manuscript the Harestua stratospheric/tropospheric BrO. We also changed the x-axis to be Time (years).**
**We are including in this discussion a plot showing only few months (Figure 5), however, we believe it does not add value to the manuscript.**

d) Fig. 8: see 6c)

**Response: **
**You mean see comment 6b. We updated old Figure 8 (now Figure 7) using orthogonal regression instead of linear regression.**

**Changed the caption of Figure 8:**
**Correlation and orthogonal regression of OMI and Harestua BrO for the data in**
**Figure 6. The legends show the mean biases and standard deviations of the**
**differences, correlation, and the orthogonal regression.**

e) Fig. 10: This plot does not allow an independent judgment whether this is a
significant signal or not. Suggested improvements:
- Increase area significantly
- Use full colorscale
- Thicker coast lines
- Align with other geospatial properties: cloud statistics, albedo, precipitation etc.

**Response:**
**Old Figure 10 (now Figure 9) has been updated: We increased the covered area,**
**used full-color scale, and added the lake line. We also added another panel to**
**show BrO enhancement over the Dead Sea Valley for September 2007.**

Further Comments
(p. 5, l. 16) "Unlike the often-used DOAS fitting method (Platt, 1994), radiances are not
ratioed to irradiances, logarithms are not taken, and no high-pass filtering is applied." I
wonder whether this is an advantage or disadvantage of the described method. What
is the intention behind this statement? My suggestion would be not to confuse the
reader and just remove it from the manuscript.

**Response:
**
**We do not see anything wrong with the above sentence. We are trying to**
**describe the algorithm and how it differs from other approaches (c.f., Platt, U.,**

"Differential optical absorption spectroscopy (DOAS)", Chem. Anal. Series, 127, 27 - 83, 1994). Describing the algorithm in detail should not confuse the readers. In our long experience with analysis of satellite spectra, the added DOAS steps do not improve the result and thus our approach has an advantage.

**Response:
**
**We do not see anything wrong with the above sentence. We are trying to describe the algorithm and how it differs from other approaches (c.f., Platt, U., "Differential optical absorption spectroscopy (DOAS)", Chem. Anal. Series, 127, 27 - 83, 1994). Describing the algorithm in detail should not confuse the readers. In our long experience with analysis of satellite spectra, the added DOAS steps do not improve the result and thus our approach has an advantage.
**

(p. 11, l.17) Why are OMI and GOME-2 data treated differently with respect to spatial averaging? Without discussion, the reader may assume that OMI data are more noisy and needed some smoothing. Please be more specific.

**Response:
**
**We have changed to use individual OMI BrO measurements without averaging. We generated new Figure 4 (in the updated manuscript) and Figure 5 (in the updated manuscript) corresponding to the use of OMI BrO data on individual pixels.**
**We see excellent agreement between OMI BrO and GOME-2 BrO with a correlation of 0.74, and a mean bias of -0.216 $\pm$ 1.13$\times$1013 molecules cm-2 (mean relative bias of -2.6 $\pm$ 22.1% ).**

(p. 13, l.21 -> l. 25) -> move to introduction

**Response: **
The text was moved to the Introduction and was changed to "Enhancement of BrO in the vicinity of salt lakes like the Dead Sea and the Great Salt Lake have been observed from ground-based measurements (Hebestreit et al., 1999; Matveev et al., 2001; Stutz et al., 2002; Tas et al., 2005; Holla et al., 2015). The active bromine compound release is due to the reaction between atmospheric oxidants with salt reservoirs. Satellite observation of salt lake BrO was first reported over the Great Salt Lake and the Dead Sea from OMI (Chance, 2006). Seasonal variations of tropospheric BrO over the Rann of Kutch salt marsh have been observed using OMI from an independent research BrO product (Hörmann et al. 2016)."

(p. 14, l. 2 -> l. 99 -> move to introduction

**Response:**
 Page 14 Line 2 to Line 9 was moved to the Introduction and was changed to "Bobrowski et al. (2003) made the first ground-based observations of BrO and SO2 abundances in the plume of the Soufrière Hills volcano (Montserrat) by multi-axis DOAS (MAX-DOAS). BrO and SO2 abundances as functions of the distance from the source were measured by MAX-DOAS in the volcanic plumes of Mt. Etna in Sicily, Italy and Villarica in Chile (Bobrowski et al., 2007). The BrO/SO2 ratio in the plume of Nyiragongo and Etna was also studied (Bobrowski et al., 2015). The first volcanic BrO measured from space was from the Ambrym volcano, measured by OMI (Chance, 2006). Theys et al. (2009) reported on GOME-2 detection of volcanic BrO emission after the Kasatochi eruption. Hörmann et al. (2013) examined GOME-2 observations of BrO slant column densities (SCDs) in the vicinity of volcanic plumes; it showed clear enhancements of BrO in 1/4 of the volcanos, and revealed large spatial differences in BrO/SO2 ratios."

[Figure]

(p.2, l. 29): "briefly analyze" characterizes an approach not suitable for a scientific article. An analysis is either profound or not scientific. In my opinion, an AMT paper should only contain profound content.

Response:

The statements "An analysis is either profound or not scientific. In my opinion, an AMT paper should only contain profound content" are truly unnecessary. The standard of our research is as high as that of anyone in the field. Our analysis is not brief and we should not have used that word. We removed it from the manuscript.

(p.3, l. 22): Please add reference documenting the OMI row anomaly:
http://projects.knmi.nl/omi/research/product/rowanomaly-background.php

**Response:
**
**The link "(http://projects.knmi.nl/omi/research/product/rowanomaly-background.php)" has been added after OMI row anomaly:**
* * *
[Figure]

**Fig. 1.** Retrievals with new SO2 cross section (Vandaele et al. (2009) . OMI orbit 30876

[Figure]

**Fig. 2.** Retrievals with new SO2 cross section (Vandaele et al. (2009)
. OMI orbit 31050

[Figure]

**Fig. 3.** Retrievals with operational SO2 cross section (Vandaele et al. (1994) OMI orbit 31050

[Figure]

BrO  05/05/2010  (OMI Orbit 30876)

**Fig. 4.** Retrievals with operational SO2 cross section (Vandaele et al. (1994) OMI orbit 30876

**Fig. 5.** Time series comparison of ground-based zenith-sky total BrO (black) at Harestua, Norway and coincident SAO OMI BrO (red) from February 2007 through August 2007.

---

## Author Response (AR3)

**Responses to the Associate Editor:**

We greatly appreciate the associate editor for his detailed comments. We are confident that by answering them the quality of the manuscript has been greatly improved. In what follows our answers are typed in blue after the associate editor comment (in black). If the text was changed in the manuscript, then the text is typed in bold blue.

For consistency, the following global changes were made:
ozone → $O_3$
$O_4$ → $O_2$-$O_2$
Figure → Fig.
el at. → *el at.*
degrees → °

Major Revision
Associate Editor Decision: Reconsider after major revisions (11 Feb 2019) by William R. Simpson
Comments to the Author:
The authors have improved the manuscript and have done supplementary calculations that were requested in the review process, which helps to understand systematic issues related to the OMBRO v3 operational retrieval. These calculations address the issues raised in the review process, but their implications are not yet integrated into the text sufficiently for this manuscript to be acceptable. Therefore, another major revision is necessary to integrate these concepts. No new calculations are being requested, but specific improvements to the text are requested.

It is important for the community to have the OMBRO v3 algorithm described in the literature, and this manuscript does that. It is also important to realize that this is an operational algorithm that should be described and can be improved, and the manuscript describes potential changes to a future version. Lastly, the manuscript needs to give caution to users that will help users to use the product properly.

Yes. We added a paragraph at the end to give caution to the users about these issues and described potential changes in future versions:

**Several important retrieval issues in the current operational algorithm that affect the quantitative use BrO VCDs have been raised in this paper such as the exclusion of $O_2$-$O_2$, nonoptimal $SO_2$ cross sections, the neglect of radiative effect of $O_3$ absorption, and the assumption of stratospheric only BrO profile. The users are advised to pay attention to these issues so that the product can be used properly. Future versions of OMBRO will include updated $SO_2$ and $O_2$-$O_2$ cross sections, corrections for the radiative transfer effect of the $O_3$ absorption and reoptimize the spectral fitting windows to mitigate the interferences of other**

**trace gases. We will also improve the AMF calculation accounting for clouds and O₃ and will consider the use of model-based climatological BrO profiles. These updates will increase the capabilities of the OMBRO retrieval to quantitatively estimate enhancements over salt lakes and in volcanic plumes.**

Main concepts:

1) Although the text now has improved error analysis and caution regarding airmass factor and spectral fitting issues, the abstract and conclusions do not reflect this addition of information. The text now says "To estimate the error associated with the AMFs calculations assuming a stratospheric profile we studied the difference between AMFs calculated using stratospheric only BrO profile and stratospheric-tropospheric profile and found that the mean absolute difference is 41%. Fig. 4 shows the AMF relative error dependency with wavelengths (bottom panel), albedo (middle panel) and VZA (top panel) as a function of the SZA. The AMFs relative errors are greater at high SZAs by about 50% for the case of wavelengths and VZAs." However, the abstract and conclusions do not convey this important issue. The abstract must be modified to indicate the issues with use of a fixed airmass factor.

We have modified the abstract to include text regarding the improved AMF error analysis. The following text has been added to the abstract:
**Additional fitting uncertainties can be caused by the interferences from O₂-O₂, and H₂CO and their correlation with BrO. AMF uncertainties are estimated to be around 10% with the used single stratospheric only BrO profile. However, under conditions of high tropospheric concentrations, AMF errors due to this assumption of profile can be as high as 50%.**

The conclusions are also essentially unaltered and need to reflect the changes made to the full text. The tracked changes mode shows that the major change to the conclusions is removal of a "roadmap" for an improved OMBRO product. That is the wrong direction in which the manuscript should be going. The manuscript should explain that these issues with respect to airmass factor, HCHO, and O2-O2 should be addressed in future retrievals. If the authors don't want to commit to making these changes in the next version (which is a choice), they can simply indicate that these issues are significant and one could consider making these improvements. The conclusions needs to express the way in which these factors affect quantitative use of the OMBRO v3 product VCDs.

We have added/modified the following in the conclusions:

**We have investigated additional fitting uncertainties caused by interferences from O₂-O₂, H₂CO, O₃, and SO₂, the impact of the choice of fitting window, the use of common mode, the orders of closure polynomials, and instrument slit functions. Uncertainties in the AMF calculations are estimated at ~10% unless the observation is made over a region with high tropospheric BrO columns. In this case, the use of a single stratospheric BrO profile is another source of uncertainty, overestimating AMFs (up to 50%) and therefore underestimating BrO VCDs.**

We summarize the retrieval issues and bring back and expand on the "roadmap" for the improved of OMBRO product at the end of the conclusion:

**Several important retrieval issues in the current operational algorithm that affect the quantitative use BrO VCDs have been raised in this paper such as the exclusion of $O_2$-$O_2$, nonoptimal $SO_2$ cross sections, the neglect of radiative effect of $O_3$ absorption, and the assumption of stratospheric only BrO profile. The users are advised to pay attention to these issues so that the product can be used properly. Future versions of OMBRO will include updated $SO_2$ and $O_2$-$O_2$ cross sections, corrections for the radiative transfer effect of the $O_3$ absorption and reoptimize the spectral fitting windows to mitigate the interferences of other trace gases. We will also improve the AMF calculation accounting for clouds and $O_3$ and will consider the use of model-based climatological BrO profiles. These updates will increase the capabilities of the OMBRO retrieval to quantitatively estimate enhancements over salt lakes and in volcanic plumes.**

2) The text on page 8 tries to describe the O4 absorption as "small", but the response to reviewer version of Figure 1 shows it is larger than BrO.
We have changed the text "$O_2$-$O_2$ has a small absorption feature" to "$O_2$-$O_2$ **has an absorption feature larger than the BrO absorption**"

Additionally, there is the band of O4 at 328nm, which is about 20% of the strength of the shown 344nm band and is in the spectral fit range. (Lampel et al., 2018 https://www.atmos-chem-phys.net/18/1671/2018/acp-18-1671-2018.pdf). Lampel et al. provide an improved cross section to the one shown in the supplement's Figure 1 that includes this second interfering peak. The authors need to show these O4 cross sections to demonstrate the potential for interference due to O4.

The supplement's version of Figure 1 (the one including O4, but also with the 328nm peak from Lampel et al.) should be used for the main text. The main text should clearly say that O4 was excluded from the OMBRO v3 fitting, and that this needs to be considered for future fitting. Cross correlations between O4 and other spectra should also be re-calculated using the Lampel et al. cross section that includes the 328nm peak. The text tries to say that "It is impossible to quantify O4...", but clearly other BrO retrievals do just that, so this text needs to be modified to be accurate. The text can still say the choice made for OMBRO v3 was to exclude it. Presumably the common mode spectrum contains at least some of this information. It also needs to be discussed that future fitting needs to consider O4 -- it is larger than BrO.

Including the O4 panel on Figure 1 means not including the mean residual. Please take the RMS of the residual and very clearly state this RMS value. To my eye, that RMS may be ~10^-3, which indicates that BrO and O4 features are marginally larger than the residual, which is typical for these retrievals.

We have updated Fig. 1 to include Lampel *et al.* (2018) $O_2$-$O_2$ cross sections and included the RMS in Fig. 1 captions: **The RMS of the fitting residuals are on the order of $9 \times 10^{-4}$, indicating that BrO spectral features are slightly bigger than typical fitting residuals**.

Additionally, we changed "It is impossible" to '**It is difficult**" added the following text to the manuscript:
**Lampel *et al.* (2018) provides spectrally resolved $O_2$-$O_2$ cross sections not only at 343 nm, but also at 328 nm (see Fig. 1) which is about 20% of the absorption at 343 nm and has not been shown in previous $O_2$-$O_2$ cross sections. Future updates to the operational OMBRO algorithm will investigate the effect of including Lampel *et al.* (2018) $O_2$-$O_2$ cross sections on the fitting.**

Comments (referring to revised version page and line numbers):

Page 4, line 9: Text says "spatial and spectral zoom modes", but spectral doesn't seem to be mentioned. I'm not actually sure that these details on the zoom mode or rebinning of the mode are really necessary for this manuscript.
The text "The spatial zoom mode (SZM) is employed every 32 days (Levelt et al., 2006): data from this mode are spatially rebinned to global-mode sampling sizes, known as the rebinned spatial zoom mode. The SZM, like the global mode (GM), has 60 cross- track pixels. These are re-binned to 30 pixels, to form "the rebinned spatial zoom mode" (RSZM) which is equivalent in pixel size to the GM data, but with reduced spatial coverage" was deleted from the manuscript.

We also removed the mention of RSZM in Section 2.2 and changed the text from "retrieved from calibrated radiance and irradiance spectra in OMI GM and RSZM level 1b data product" to "**retrieved from calibrated OMI radiance and irradiance spectra**"

Page 6, line 2: The text indicates that row anomaly flags are carried forward, but is not fully clear on how they are used in the retrieval. Presumably these pixels are not used, in which case the text could just say that.
The following text was added to the manuscript
"**We recommend not to use pixels affected by the row anomaly despite being processed by the retrieval algorithm.**"

Page 6, line 13: The sentence about DOAS does not always reflect modern usage of the method. Since this text is describing the method used for OMBRO, there is no need to discuss DOAS and this sentence should be cut from the text.
The sentence was removed from the text.

Page 6, lines 20-27: This section is not clear. Seemingly two irradiance spectra are being discussed, a high resolution one for wavelength calibration, and one at OMI resolution for analysis using equation 1, presumably I0, and if that is, please be clearer. What is the principal component analysis? Is this irradiance spectrum used as the solar reference for all years after 2007?

Principal component analysis (PCA) is a statistical procedure to derive the empirical orthogonal functions or principal components from individual observations. We performed PCA for each cross-track position separately using three years of individual solar irradiance spectra normalized

to 1 AU Sun-Earth distance. Yes. The first principal component of solar irradiance at each cross-track position is used as the solar reference throughout the entire OMI record.

To make it clear, we modified this sentence as follows and have moved it to the paragraph below Eq. 1:
**To improve cross-track stripe biases (Section 3.4), the OMI daily solar irradiance ($I_0$) is substituted by the first principal component of the solar irradiances measured by OMI between 2005 and 2007 (one for each cross-track position). The principal component derived between 2005 and 2007 is used to process the entire mission.**

And the sentence after this sentence stays at the same place with a small change: "Radiance wavelength calibration is performed for a representative **swath** line of radiance measurements (usually in the middle of the orbit) to determine a common wavelength grid for reference spectra.**"**

Page 7, lines 5-8: This section talks about "scan lines", "60 cross-track pixels", and "100 across-track swath lines". It seems like the language could be made more clear. Figures in the response to reviewer clearly show the dimensions as "cross track index" and "swath line index" (along flight direction). Please use consistent language or explain how these terms are different.

To make it clearer and use consistent language, "scan lines" was changed to "**swath lines**" and the paragraph was rewritten as:
**To process each OMI orbit it is split into blocks of 100 swath lines. Spectral fitting is then performed for each block by processing the 60 cross-track pixels included in each swath line sequentially before advancing to the next swath line. The spectra are modeled according to Eq. (1):**

Page 7, line 13: Presumably the approximate Gaussian spectral slit function (0.42 nm FWHM) was used for convolution. Please be clear on what was used.
No. The approximate Gaussian slit function with 0.42 nm FWHM is only used for plotting Figs. 1 and 12.
We modified the sentence so now it reads "**for each cross-track position from original high-resolution cross sections convolved with the corresponding pre-launch OMI slit functions (Dirksen et al., 2006)**"

Page 8, lines 4-7: It appears that the index j is spectral and k is cross track position on the detector. Making this index naming more clear can help in clarifying some of this text, such as "An initial fit of several hundred pixels per cross-track position determines the common mode spectra (one spectrum per cross-track position, between 30 N and 30S) as the average of the fitting residuals." The word "pixels" is probably along-track position. Are all of the along-track positions between 30N and 30S used, or are "several hundred" used? Please reword to be clearer and to make use of the indices.

Actually, the index $j$ is not spectral and $k$ is not cross track position on the detector. Instead, the $i$, $j$, and $k$ indices indicate contributions at different stages of the spectra modelling (before Beer-

Lambert law, Beer-Lambert law and after Beer-Lambert law for *i*, *j*, and *k* respectively) as shown in Eq. 1 and the text below.

This section was reworded as:
**For each cross-track position, an initial fit of all the pixels along the track between 30ºN and 30ºS is performed to determine the common mode spectra, derived as the average of the fitting residuals.**

Please note that while we were reviewing Eq. 1 we discovered that *j* is not a subscript of $\beta$ so the equation was updated so that $\beta j$ is $\beta_j$.

Page 9, line 20: It should be noted that a fixed profile is also inconsistent with the varying tropopause height (both with latitude and dynamically e.g. Salawitch et al. 2010)
Yes, we added the sentence to the manuscript:
**It should be noted that a fixed profile is inconsistent with the varying tropopause height (both with latitude and dynamically e.g. Salawitch *et al*. 2010) and therefore with the profile shape in the stratosphere, but the impact on the AMF is typically small as the scattering weight does not change much in the stratosphere.**

Page 12, line 19: The percent symbol is mis-placed in what should be 50%.
Changed %50 to **50%**.

Page 13, line 20: Re-word this sentence to say "GOME-2 has a descending orbit..." instead of the awkward "..for GOME-2" and "...for OMI."
Page 13, line 20 was changed to: **GOME-2 has descending orbit with a local equator crossing time (ECT) of 9:30 am and OMI has ascending orbit with an ECT of 1:45 pm.**

Page 14, line 13: Give these percentages also in molecule / cm^2 units (for consistency with other text).
The percentages have also been added in units of molecule cm$^{-2}$.
That sentence now reads as:
**Our study shows that OMI has negative mean biases of 0.35×10$^{13}$ molecules cm$^{-2}$ (12%), 0.33×10$^{13}$ molecules cm$^{-2}$ (10%), 0.25×10$^{13}$ molecules cm$^{-2}$ (17%), and 0.30×10$^{13}$ molecules cm$^{-2}$ (10%) for Alaska, Southern Pacific, Hudson Bay, and Greenland, respectively.**

Page 15, line 2: Amundsen is an Icebreaker that was in the Canadian Arctic Ocean.
The manuscript was changed to:
**made from the Icebreaker Amundsen, in the Canadian Arctic Ocean**

Page 15, line 10: The "number of negative pixels" needs a reference (e.g. out of N total pixels).
The total number of total pixels was added to the manuscript in Table 2 caption. The caption now reads:
**Error analysis studies. For reference, the total number of retrieved pixels not affected by the row anomaly is 58112.**

The first paragraph of Section 4.2 including this sentence was removed. We have addressed reviewer #3 comments regarding the differences between V2 and V3 of the OMBRO in Table 2. The comparison of V2 and V3 in Fig. 10 is not needed and also it does not fit into the current structure of the manuscript.

Page 15, line 13: This says V9 for the new retrieval -- V3?
Yes, V3. However, the entire paragraph is removed as explained above.

Page 15, line 27: Give months for SH winter for clarity. This whole section is fairly long for a simple concept that the distributions look like other satellite retrieved seasonal BrO patterns.
This paragraph was reworded as:
**The seasonality is different between northern and southern hemispheres. In the northern hemisphere, BrO values are larger in spring (March) with widespread enhancement and smaller in fall (September/December). In the southern hemisphere, BrO values are larger in southern hemispheric spring and summer (i.e., September and December) and smaller in the winter (June).**

Page 16, line 12: The Hörmann et al. (2016) paper indicates that OMI spectra were analyzed, not that the OMBRO product was used. I believe they use OMI data but not OMBRO. Please be clearer.
The OMBRO is used by us in this paper. To clarify, we have modified the sentence to below:
"**Following recent work by Hörmann et al. (2016) over the Rann of Kutch using OMI BrO retrievals from an independent research product, we have explored the capability of our OMBRO product to observe similar enhancements in other salt lakes**"

Page 16, line 19: Figure 3 is cited, but it should be Figure 3, left panel (the profile that was used), rather than the right panel, which was not used in OMBRO v3, but only for sensitivity studies.
(Figure 3) was changed to **(Fig. 3 left panel)**

Page 16, lines 25-29: Do not mention results of this other paper unless you intend to show it here. Specifically, you cannot say "around 4.5 x 10^13 molecules cm^-2" unless you want to show that information here.
The following text was removed from the manuscript: We have done a preliminary analysis of salt water bodies including the Rann of Kutch. Although this work is not fully complete and will be a separate paper, however, we see maximum BrO VCDs appearing during March–May every year from 2004 – 2015 similar to what was reported by Hörmann et al., 2016. The BrO VCDs we see are around 4.5 x $10^{13}$ molecules cm$^{-2}$.

Page 17, line 25: The unit "molecules cm^-2" is missing; add it.
Added **molecules cm$^{-2}$**

Page 18, line 5: Add "potentially" before "real BrO". This point needs study, and you are just showing that the old SO2 cross sections were bad. It would be a different study to try to quantify the BrO, which you don't attempt here, and would be difficult due to AMF issues.
Added **potentially**

Page 19, line 9: Note here that the SNO observations only occur in certain regions of the globe.
Added "**(SNOs), which only occur around 75ºN and 75ºS**"

Page 19, line 14: replace "no much" with "little"
Replaced "no much" with "**little**"

Page 19, line 23: Need to discuss that more work is needed to address AMF and spectral issues with these case studies to turn the qualitative identification of BrO into quantitative measurements of BrO VCD.
A new paragraph was added to the conclusions.
**Future versions of OMBRO will include updated $SO_2$ and $O_2$-$O_2$ cross sections, corrections for the radiative transfer effect of the $O_3$ absorption and reoptimize the spectral fitting windows to mitigate the interferences of other trace gases. We will also improve the AMF calculation accounting for clouds and $O_3$ and consider the use of model-based climatological BrO profiles. These updates will increase the capabilities of the OMBRO retrieval to quantitatively estimate enhancements over salt lakes and in volcanic plumes.**

Page 34, Table 2: For V3 algorithm, list wavelengths for consistency. List the total number of observations so that we can tell what the "number of negatives" is in reference to. To my eye, with O2-O2 looks pretty similar to V3 in terms of median VCD, uncertainty, and number of negatives.
The total number of pixels was added to the captions of Table 2 and wavelength was added to V3. We updated the values of Table 2 since we realized that pixels affected by row anomaly were not filtered out previously. After filtering out row anomaly pixels the median from the fitting including $O_2$-$O_2$ is 12% lower.

Figure 1: The text uses molecules cm^-2 fairly consistency, but this figure lists DU and mDU. I think that all (except O3) should be converted to molecules cm^-2.
Figure 1 has been updated to show the values in units of molecules $cm^{-2}$.

Replace the bottom right panel with the O4 spectrum. Clearly indicate the typical RMS on the figures, so that the vertical scales can be compared to typical RMS.
Figure 1 has been updated and now O4 spectrum from Lampel et al. (2018) is shown in the bottom right panel. Typical RMS is mentioned in the figure caption.
The caption now reads:
**Cross sections used in the current operational BrO algorithm except for the $SO_2$ cross section at 298 K which is to be used in the next version. The black lines are the original cross sections, the color lines show the cross sections convolved with approximate OMI slit function (which is assumed to be a Gaussian with 0.42nm full width at half maximum). The $O_2$-$O_2$ calculation is based on Lampel *et al.* (2018) cross sections. The BrO cross section after multiplication with the wavelength-dependent AMFs used these parameters for the AMF calculation: albedo = 0.05, SZA= 5.0 º, and VZA = 2.5 º). The RMS of the fitting residuals are on the order of $9x10^{-4}$, indicating that BrO spectral features are slighly bigger than typical fitting residuals.**

Figure 3: This figure needs to be clear that the left panel is used in OMBRO V3, while the right panel was used for a quick test.
Figure caption was updated to read:
**(Left) A mostly stratospheric vertical BrO profile used for air mass factors calculations in OMBRO V3. Total BrO, BrO < 15 km, BrO < 10 km, and BrO < 5km are $2.05 \times 10^{13}$, $5.06 \times 10^{12}$, $1.55 \times 10^{12}$, and $2.87 \times 10^{11}$, respectively. (Right) A stratospheric tropospheric vertical BrO profile used to investigate the impact of high tropospheric BrO columns on air mass factors calculations. Total BrO, BrO < 15 km, BrO < 10 km, and BrO < 5km are $6.99 \times 10^{13}$, $5.45 \times 10^{13}$, $5.10 \times 10^{13}$, and $4.97 \times 10^{13}$, respectively.**

Figure 4: Other places use percent errors, so please change the scale bar to percent.
Figure 4 has been updated so the scale bar is in %.
Figure caption now reads:
**The percentage of relative AMF errors as a function of the SZA and the wavelength (top panel), albedo (middle panel) and VZA (bottom panel) when using the stratospheric only BrO profile (Fig. 3, left panel) in the case there exists a significant tropospheric BrO column as shown in the stratospheric-tropospheric BrO profile (Fig. 3, right panel).**

Figure 5: Note the range of latitudes where these SNOs happen.
The figure caption now reads:
**Time series comparison of SAO OMI (red) BrO and BIRA GOME-2 (blue) BrO VCDs from February 2007 to November 2008 using simultaneous nadir overpasses occurring at high latitudes, around 75º S/N, and within 2 minutes between OMI and GOME-2 observations.**

Figure 6: Put units on the bias and stdev of differences and clarify what they are on the figure.
Figure 9: Same fixes as Figure 6

The units are consistent with those on the x- and y-axis, therefore we think they are not necessary inside the plot where they will contribute to overcrowding the figure. The reporting of the bias, standard, correlation and the orthogonal regression line has been modified to identify each one of them.

Figure 13: Note that the straight lines are borders of the state of Utah, and the curving lines represent the Great Salt Lake (Eastern oval area) and Salt flats (Western oval area).
We assume the editor means Figure 12.
The following was added to the Figure 12 caption (now Figure 11): **The straight lines are borders of the state of Utah, and the curving lines represent the Great Salt Lake (Eastern oval area) and Salt flats (Western oval area).**

Figure 14: Add black boxes outlining each of the four panels, which will make it easier to see that there are four panels.

Black boxes outlining each of the four panels have been added to Figure 14 (now Figure 13).

Additional updates:

To improve the readability of the manuscript and reduce some redundancy we have done some modifications by moving some sentences around without changing the meaning and adding additional text to provide better description of the error analysis.

The AMF errors due to the use of stratospheric only BrO profile are discussed in 2[nd] paragraph of Section 3.3 and 4[th] paragraph of Section 3.5. Now it is discussed only in Section 3.5.

In Section 3.3, it now reads: For conditions with enhanced BrO in the lower troposphere, using this profile will overestimate the AMFs and therefore underestimate the BrO VCDs **as discussed in Section 3.5**.

In the 4[th] paragraph of Section 3.5, it now reads:
**To estimate the AMF error associated with enhanced tropospheric concentrations we have studied the difference between AMFs calculated using the stratospheric only BrO profile and a stratospheric-tropospheric profile as shown in the right panel of Fig. 3. Fig. 4 shows the dependency of the relative AMF difference with respect to wavelength (bottom panel), albedo (middle panel) and VZA (top panel) as a function of the SZA between calculations performed using these two profiles. The use of stratospheric only BrO profile can lead to AMF errors up to 50% depending on albedo and viewing geometry. On average, using the stratospheric only BrO profile overestimates AMF and underestimates VCD by 41%.**

Also, the two paragraphs about retrieval sensitivity analysis have been reorganized:

**We have performed sensitivity analysis of OMI BrO VCD with respect to various retrieval settings using orbit 26564 on 13 July 2009. Table 2 shows the median VCDs, median fitting uncertainties and the number of negative VCD pixels for each configuration. Table 3 summarizes the overall fitting error budget including the random fitting uncertainty, cross sections errors (as reported in the literature), and various retrieval settings. We studied five wavelength windows including the current operational window (319.0-347.5 nm) version 2 window (323.0-353.5 nm), version 1 (340.0-357.5 nm) and two extra windows exploring the impact of extending the window to shorter wavelengths (310.0-357.5 nm) and reducing it by limiting its extension to wavelengths above 325 nm (325.0-357.5). The choice of fitting window can cause significant differences in BrO VCDs of up to 50%. The current window results in the most stable retrievals with the smallest number of pixels with negative VCD values.**

**Including the interference of $O_2$-$O_2$ leads to a decrease of the median VCD by ~12% and an increase of the median fitting uncertainty by ~10% with respect to the operational set up. Excluding $H_2CO$ from the fitting significantly reduces the retrieved BrO columns by ~37%, given that the strong anticorrelation between both molecules is not taken into account. Fitting the mean residual (common mode) has a small impact in the retrieval results, the**

**median VCD only changes ~3%, but reduces the median fitting uncertainty by ~30% with respect to the exclusion of the common mode. To study the impact of the slit functions we have performed the retrieval using both online slit functions, modelled as a Gaussian, and the preflight instrument slit functions. The median difference between these two retrievals is 27% for orbit number 26564. We have investigated the impacts of the order of scaling and baseline polynomials; it can cause uncertainties of ~10% as shown in Table 3.**

In Section 4.1, the sentence "Due to different orbits, all these SNOs occur at high latitudes around 75ºS/N" is changed to "**Given Aura and Metop-A satellite orbits, all these SNOs occur at high latitudes around 75ºS/N**".

We added the following sentence to the 3rd paragraph of the conclusion section:
**Monthly mean OMBRO VCDs during 2007-2009 show negative biases of 0.25-0.35×10$^{13}$ molecules cm$^{-2}$ (10-17%) over Alaska, Southern Pacific, Hudson Bay, and Greenland.**

and slightly modified the following:
**The BrO seasonality in Harestua is well captured by the OMI BrO and OMBRO retrieval showing a reasonable good agreement with the ground-based measurements. The correlation between both datasets is of 0.46 and the mean bias 0.12±0.76×10$^{13}$ molecules cm$^{-2}$ (mean relative bias of 3.18±16.30%).**

We added the following sentence to the 4th paragraph of the conclusion section:
**Finally, we have explored the feasibility of detecting enhanced BrO column over salt lakes and in volcanic plumes using OMBRO retrievals. We found enhancement of the BrO with respect to the background levels of 5-10×10$^{12}$ molecules cm$^{-2}$ over the U.S. Great Salt Lake.**

As the previous Fig. 10 was removed, the previous figures after that have been updated.
Fig. 11 becomes Fig 10
Fig. 12 becomes Fig 11
Fig. 13 becomes Fig 12
Fig. 14 becomes Fig 13

[revised manuscript text omitted]

**Figures and Figure Captions**

[revised manuscript text omitted]

The difference in the mean of the VCDs retrieved using the different fitting windows are always smaller than %50.

Additional sensitivity studies also shown in Table 2 include excluding from the fitting interfering molecules ($O_4$ and $CH_2O$), using pre-flight measurements of the slit function or calculating them for each orbit, not including the mean residual (common mode) in the spectral fitting, and changing the order of the closure polynomials. In these experiments, everything else is kept the same as in the operational retrieval. In Table 2, we list the median VCDs and the median uncertainties for BrO for 13 July 2009 orbit number 26564 for each one of these retrieval configurations except for the test of the different orders of the closure polynomials. The results of the polynomial sensitivity test are summarized in Table 3.

| Page 15: [3] Deleted | Raid M Suleiman | 3/7/19 4:39:00 PM |
|---|---|---|

Table 2 shows the median VCD, median uncertainties and the number of negative pixels for the current operational version (V3), V2, and V1. Fig. 10 shows the monthly mean averages for V3 and V3 for the months of February and May of 2008. The differences on the VCD are about 30% in V3 comparing to V2. In comparison with V2 retrieval, the new retrieval (V9) does not show a large increase in the VCD concentrations, especially at the north polar region. The BrO background concentrations over the Pacific Ocean remain the same between the two versions, however, there are more retrieved VCDs.

| Page 16: [4] Deleted | Raid M Suleiman | 2/20/19 11:26:00 AM |
|---|---|---|

We have done a preliminary analysis of salt water bodies including the Rann of Kutch. Although this work is not fully complete and will be a separate paper, however, we see maximum BrO VCDs appearing during March–May every year from 2004 – 2015 similar to what was reported by Hörmann et al., 2016. The BrO VCDs we see are around 4.5 x $10^{13}$ molecules cm$^{-2}$.

| Page 38: [5] Deleted | Raid M Suleiman | 2/22/19 3:15:00 PM |
|---|---|---|

[Figure]

[Figure]

[Figure]

| Page 38: [6] Deleted | Microsoft Office User | 3/9/19 8:31:00 AM |
|---|---|---|

s

| Page 38: [6] Deleted | Microsoft Office User | 3/9/19 8:31:00 AM |
|---|---|---|

s

| Page 38: [7] Deleted | Raid M Suleiman | 2/22/19 3:16:00 PM |
|---|---|---|

**bottom**

| Page 38: [7] Deleted | Raid M Suleiman | 2/22/19 3:16:00 PM |
|---|---|---|

**bottom**

| Page 38: [7] Deleted | Raid M Suleiman | 2/22/19 3:16:00 PM |
|---|---|---|

**bottom**

| Page 38: [8] Formatted | Raid M Suleiman | 3/7/19 8:05:00 PM |
|---|---|---|

Font:Not Bold, Complex Script Font: Not Bold, English (US)

| Page 38: [8] Formatted | Raid M Suleiman | 3/7/19 8:05:00 PM |
|---|---|---|

Font:Not Bold, Complex Script Font: Not Bold, English (US)

| Page 41: [9] Deleted | Raid M Suleiman | 3/7/19 8:13:00 PM |
|---|---|---|

[Figure]

Figure 10. Monthly averages for February and May 2008 for version 3 and version 2.